# Montessori-Instruct: Generate Influential Training Data Tailored for Student Learning

**Xiaochuan Li**♦*, **Zichun Yu**♦, **Chenyan Xiong**♦
♦School of Software, Tsinghua University
♦Language Technologies Institute, Carnegie Mellon University
`li-xc20@mails.tsinghua.edu.cn`
`zichunyu@andrew.cmu.edu`
`cx@cs.cmu.edu`

## Abstract

Synthetic data has been widely used to train large language models, but their generative nature inevitably introduces noisy, non-informative, and misleading learning signals. In this paper, we propose Montessori-Instruct, a novel data synthesis framework that tailors the data synthesis ability of the teacher language model toward the student language model's learning process. Specifically, we utilize local data influence of synthetic training data points on students to characterize students' learning preferences. Then, we train the teacher model with Direct Preference Optimization (DPO) to generate synthetic data tailored toward student learning preferences. Experiments with Llama3-8B-Instruct (teacher) and Llama3-8B (student) on Alpaca Eval and MT-Bench demonstrate that Montessori-Instruct significantly outperforms standard synthesis methods by 18.35% and 46.24% relatively. Our method also beats data synthesized by a stronger teacher model, GPT-4o. Further analysis confirms the benefits of teacher's learning to generate more influential training data in the student's improved learning, the advantages of local data influence in accurately measuring student preferences, and the robustness of Montessori-Instruct across different student models. Our code and data are open-sourced at `https://github.com/cxcscmu/Montessori-Instruct`.

## 1 Introduction

Synthetic training data is highly effective in various applications of large language models (LLMs) (Lu et al., 2023), spanning from general pretraining (Allal et al., 2024; Zhou et al., 2024), instruction-tuning (Tong et al., 2024) to domain-specific scenarios such as mathematics (Yu et al., 2023) and coding (Jiang et al., 2024). The advantages of synthetic data include its low cost, convenience, and flexibility, making them an appealing choice for scaling up training data (Yue et al., 2024), mitigating the shortage of human labels (Chang et al., 2024), and improving data diversity (Sun et al., 2023).

Typical data synthesis methods (Wang et al., 2023) employ an instruction-tuned teacher model and prompt it with seed data to generate synthetic training data for a student model. It is widely observed that the teacher-generated data can be noisy and non-informative (Bauer et al., 2024), their simple and uniform format may lead to pattern overfitting (Chen et al., 2024), and their biased and ungrounded content can introduce ambiguity in AI alignment (Liu et al., 2024). These are fundamental challenges of synthetic data as they can mislead students and sometimes even result in model collapse (Shumailov et al., 2023a; Seddik et al., 2024).

In this paper, we propose Montessori-Instruct, a novel data synthesis framework designed to generate more tailored and informative data by directly optimizing the synthesis ability of the teacher toward the student's learning preferences. We first leverage influence functions (Koh & Liang, 2017; Yu et al., 2024b) to precisely measure the utility of synthetic data–its ability to effectively train the students. Then, we optimize the parameters of the teacher model according to the student's preferences through Direct Preference Optimization (DPO) (Rafailov et al., 2024). The preference-optimized teacher then synthesizes influential training data for the students. As shown in Figure 1,

---
*Part of this work is done while visiting CMU.

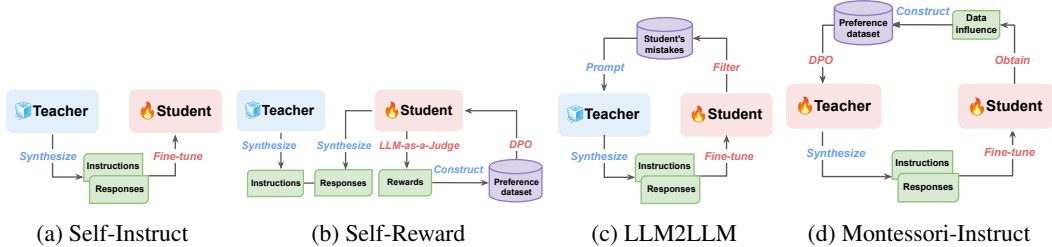

Figure 1: Data synthesis methods with standard teacher (data synthesizer) and student (target) setups.

rather than employing LLM-as-a-judge (Zheng et al., 2024) to evaluate and filter data by quality (Yuan et al., 2024) or prompting teachers to generate harder examples (Lee et al., 2024) Montessori-Instruct directly optimizes the teacher according to students' learning preferences, leading to more customized, flexible, and effective synthetic training data for the students.

Our experiments use Montessori-Instruct to synthesize 10K instruction-response pairs with Llama3-8B-Instruct (Meta, 2024) as teacher and train Llama3-8B/Tinyllama-1.1B (Zhang et al., 2024) as students. The results show that Montessori-Instruct achieves relative improvements of 18.35% and 46.24% over Self-Instruct on in-domain Alpaca Eval (Dubois et al., 2024) and out-of-domain MT-Bench (Zheng et al., 2024), respectively. The benefits of Montessori-Instruct are more pronounced compared to state-of-the-art data synthesis methods such as Self-Reward and LLM2LLM, as well as data synthesized by the cutting-edge LLM, GPT-4o (OpenAI, 2024). The results on a wide range of general NLP tasks (e.g., MMLU (Hendrycks et al., 2020) and GSM8K (Cobbe et al., 2021)) further demonstrate the generalization capabilities of Montessori-Instruct.

Further analyses reveal a strong correlation between the teacher's optimization process and the student's performance, demonstrating that Montessori-Instruct enables the teacher to generate data aligned with students' preferences to enhance its learning. Ablation studies highlight the advantages of using data influence to reflect students' preferences, the effectiveness of optimizing the teacher parameters over solely bootstrapping the data, and the robustness of Montessori-Instruct across different seed data, multiple iterations, and a variety of student models.

Our main contributions are summarized as follows:

1. We propose Montessori-Instruct, a novel data synthesis framework that tailors the data synthesis ability of the teacher toward the student's learning.

2. We incorporate influence functions to accurately capture the student's data preferences and effectively guide the teacher's optimization directions.

3. Our empirical results demonstrate the effectiveness and robustness of Montessori-Instruct in improving students' learning outcomes by tailoring synthetic data generation to align with student learning preferences.

## 2 RELATED WORK

Synthetic data has been shown highly effective in various applications of large language models (Lu et al., 2023), including pretraining (Allal et al., 2024; Zhou et al., 2024), instruction-tuning (Tong et al., 2024; Yue et al., 2024), mathematics (Yu et al., 2023) and coding (Jiang et al., 2024). Typical approaches like Self-Instruct (Wang et al., 2023) leverages an instruction-tuned teacher to generate instruction-response pairs given a small amount of seed data. Following the similar pipeline, Self-Guide (Zhao et al., 2024) and Self-Alignment (Sun et al., 2023; Guo et al., 2024) further enhance data quality for specific tasks, such as safety, truthfulness, and instruction-following, by carefully curating task-relevant seeds. In parallel, Instruction Backtranslation (Li et al., 2023) and Bonito (Nayak et al., 2024) collect massive texts from the internet as responses, prompt LLMs to synthesize instructions reversely, and select high-quality candidates.

Despite its promising potential, synthetic data primarily rely on the teacher's free-form generations, thus is inevitably often biased, non-informative, and misleading (Bauer et al., 2024; Liu et al., 2024).

The discrepancy between synthetic data and real-world sources often results in a misalignment with human values and preferences (Liu et al., 2024), raising the risk of training student models that are biased (Feng et al., 2023; Liu et al., 2021), ungrounded (Liu et al., 2022; Patel & Pavlick, 2022), or misrepresentative of real-world scenarios (Ji et al., 2023; Hu et al., 2024b). It is also observed that task-specific synthetic data often lacks diversity (Yu et al., 2024a), whereas general synthetic data suffers from pattern overfitting (Chen et al., 2024) and the memorization of the synthesis model's training data (Van Breugel et al., 2023). Another challenge of synthetic data training is the phenomenon of model collapse (Shu et al., 2023), where the massive noise in unregulated synthetic data leads to the disappearance of the tails of the original content distribution and ineffective student models (Seddik et al., 2024).

To address these limitations, researchers have explored various approaches to improve the utility of synthetic data (Shu et al., 2023; Wang et al., 2024). One line of work focuses on filtering out noisy synthetic data, using techniques like ranking synthetic data with an additional reward model (Shu et al., 2023), verifying the truthfulness of responses via programs (Dong et al., 2024), prompting LLMs to judge the data quality (Zheng et al., 2024), and ensemble of multiple teacher (Lee et al., 2023). One can also directly adjust the teacher's synthesis strategies to generate more useful data for students (Lee et al., 2024; Yuan et al., 2024). For instance, LLM2LLM (Lee et al., 2024) collects data points that the student answers incorrectly and prompts the teacher to bootstrap similar data, thereby generating targeted data to strengthen the student's weaknesses. Another potential path, such as Self-Reward (Yuan et al., 2024), is to employ LLM-as-a-judge (Zheng et al., 2024) to assign each response a discrete reward score and optimize the student to generate highly rewarding responses.

The last body of related work is data influence functions (Hampel, 1974), a commonly used technique for measuring the utility of data on a model's performance. Influence function (Hampel, 1974; Koh & Liang, 2017; Bae et al., 2022) quantifies the change in reference loss when a data point is upweighted in the training set (Koh & Liang, 2017). It often serves as a theoretical tool to analyze data utility (Choe et al., 2024) and attribute model behavior (Park et al., 2023). Recent work has applied influence functions to facilitate model-aware data selection in pretraining or instruction-tuning, using first-order approximation (Xia et al., 2024), linear datamodels (Engstrom et al., 2024), and data influence models (Yu et al., 2024b). These methods have been shown to be more effective than traditional rule-based techniques in data selection, mostly notably in the pretraining stage (Engstrom et al., 2024; Yu et al., 2024b).

## 3 MONTESSORI-INSTRUCT

This section first introduces the overall framework of MONTESSORI-INSTRUCT (§ 3.1) and then elaborates its two main components: local data influence collection (§ 3.2) and student-preference-guided teacher optimization (§ 3.3).

### 3.1 OVERALL FRAMEWORK

Standard data synthesis methods (Wang et al., 2023; Yuan et al., 2024; Lee et al., 2024) begin with a teacher model $\mathcal{M}$ and a seed prompt $p$ formed using a few-shot sample of example data. The teacher model processes the seed $p$ to generate a set of $N$ new instructions, $\{x_i \mid 1 \leq i \leq N\}$, that follow a similar format to the seed but with a variety of contents. Each generated instruction $x_i$ is then used to prompt the teacher to synthesize the corresponding response $y_i$. This yields a set of instruction-response pairs $\{(x_i, y_i) \mid 1 \leq i \leq N\}$ that are then used to train the student model $m$.

Montessori-Instruct upgrades this standard data synthesis pipeline with the optimization of the teacher model toward the student's learning preferences. The student-preference-guided teacher optimization starts with prompting the teacher to generate a *probing dataset* $\mathcal{D}_{\text{probing}}$ using Self-Instruct and then collecting these data points' local data influence $\mathcal{I}_m$ on the student model (§ 3.2). The collected data preferences form the *preference dataset* $\mathcal{D}_{\text{preference}}$, and Montessori-Instruct uses it to update the teacher model via Direct Preference Optimization (DPO) (Rafailov et al., 2024) (§ 3.3). The optimized teacher then generates the actual training dataset to train the student model $m$. The process can be iterated multiple rounds to continually refine the teacher according to the student's updated preferences. This process is illustrated in Figure 2 and discussed in detail in the next two sections.

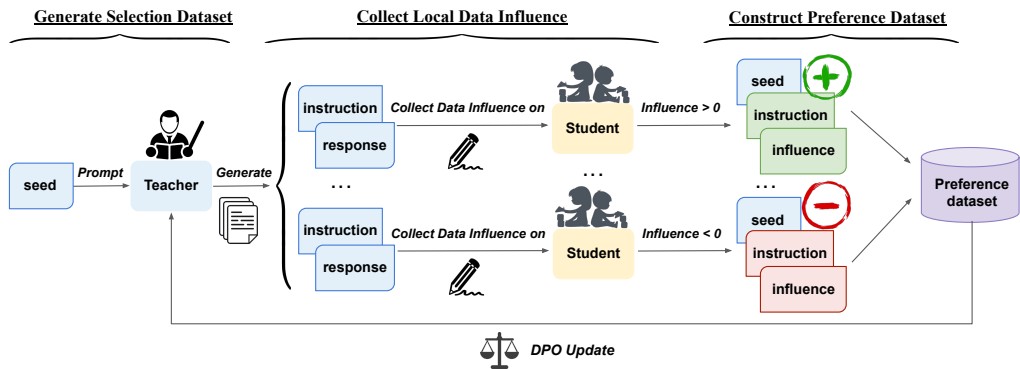

Figure 2: Student-Preference-Guided teacher optimization in Montessori-Instruct.

## 3.2 LOCAL DATA INFLUENCE COLLECTION

A key component of our framework is to precisely measure the utility of synthetic data, i.e., how good they are at improving the student's learning outcomes. This question is often approached using influence functions (Weisberg & Cook, 1982; Koh & Liang, 2017), which was designed to quantify changes in reference loss when a data point $(x_i, y_i)$ is upweighted in the training sets Park et al. (2023), thus reflecting the utility of this data point to the student's learning.

In order to efficiently calculate the data influence, we follow Yu et al. (2024b) and approximate influence functions locally, using the change of the model's reference loss before and after training on a single data point $(x_i, y_i)$:

$$\mathcal{I}_m(x_i; \mathcal{D}_{\text{ref}}) \approx -\mathcal{L}(\mathcal{D}_{\text{ref}} \mid \mathcal{A}(y_i \mid x_i; m)) + \mathcal{L}(\mathcal{D}_{\text{ref}} \mid m), \tag{1}$$

$$\text{where } \mathcal{L}(\mathcal{D}_{\text{ref}} \mid m) = \mathbb{E}_{(x,y) \sim \mathcal{D}_{\text{ref}}} \ell(y \mid x; m), \tag{2}$$

where $\mathcal{D}_{\text{ref}}$ denotes the reference data that measure the student's capability, and $\ell(y|x; m)$ is the loss of student $m$ on an input-output pair $(x, y)$. $\mathcal{A}(y_i \mid x_i; m)$ refers to the optimization operation of student $m$ on data $(x_i, y_i)$, e.g., one-step training with Adam (Kingma & Ba, 2015) on $(x_i, y_i)$.

The local data influence, $\mathcal{I}_m(x_i; \mathcal{D}_{\text{ref}})$, represents how the instruction-response pair $(x_i, y_i)$ impacts the student's learning outcome as measured on the reference data. A positive $\mathcal{I}_m$ indicates that the data benefits the student's reference performance, while a negative $\mathcal{I}_m$ shows the opposite. A complete theoretical derivation of local data influence is provided in Appendix B.

## 3.3 STUDENT-PREFERENCE-GUIDED TEACHER OPTIMIZATION

After calculating local data influence for each instruction in the probing dataset $\mathcal{D}_{\text{probing}}$, we pair every two instructions with positive and negative influence, along with their corresponding seed prompt $p$, to construct the preference dataset:

$$\mathcal{D}_{\text{preference}} = \{(p, x^+, x^-) \mid \mathcal{I}_m(x^-; \mathcal{D}_{\text{ref}}) < 0 < \mathcal{I}_m(x^+; \mathcal{D}_{\text{ref}})\}. \tag{3}$$

We then apply DPO to optimize the teacher model $\mathcal{M}$ toward the student's learning preferences:

$$\mathcal{L}_{\text{DPO}}(\mathcal{M}^*; \mathcal{M}) = -\mathbb{E}_{(p, x^+, x^-) \sim \mathcal{D}_{\text{preference}}} [\log \sigma(\beta \log \frac{\mathcal{M}^*(x^+ \mid p)}{\mathcal{M}(x^+ \mid p)} - \beta \log \frac{\mathcal{M}^*(x^- \mid p)}{\mathcal{M}(x^- \mid p)})], \tag{4}$$

where $\beta$ is a parameter that controls the deviation from the initial teacher $\mathcal{M}$ and $\sigma$ is the logistic function. The updated teacher, $\mathcal{M}^*$, after one or multiple iterations, is then used to synthesize the training data for the student model $m$.

## 4 EXPERIMENTAL METHODOLOGIES

This section details our main experimental setups, including a thorough configuration of the data synthesis process, the chosen baselines, and the evaluation methods.

**Data Synthesis Process.** We choose Llama3-8B-Instruct (Meta, 2024) as the teacher, and train Llama3-8B (Meta, 2024) and Tinyllama-1.1B (Zhang et al., 2024) as students. We merge the text in instruction and input fields of Alpaca GPT-4 dataset (Taori et al., 2023), consisting of 52K entries, to create our seed pool. We follow the 8-shot seed proposed in Self-Instruct (Wang et al., 2023) to prompt the teacher to generate instructions, with 6 out of the 8 randomly sampled from the seed pool and 2 sampled from the synthetic instructions in the teacher's previous iterations. Detailed prompts are provided in Figure 13.

Following Yuan et al. (2024), we initially use the unoptimized teacher model to synthesize 1K data to warm up the student. Then, we generate 4 instructions for each seed and 1 response for each instruction and filter out similar instructions whose Rough-L score exceeds 0.7, resulting in a probing dataset of 10K prompt-instruction-response triplets. For each instruction-response pair in the probing dataset, we collect local data influence using the loss difference of the student model on the reference data (Alpaca GPT-4) before and after one-step training. Then, we construct a preference dataset comprising 6,792 entries, where each entry represents a seed-instruction pair with positive and negative influences. This preference dataset is used to train the teacher with Direct Preference Optimization (DPO) (Rafailov et al., 2024). Finally, we use the optimized teacher to synthesize 10K data to train the student from scratch. In the subsequent iterations, we optimize the teacher using similar steps, but with the updated student from last iteration to collect data influence. For both the teacher and student training, we utilize AdamW optimizer (Loshchilov & Hutter, 2019) along with WSD scheduler (Hu et al., 2024a). Both models are trained for one epoch. For teacher's generation, we use vLLM (Kwon et al., 2023) as our decoding engine and provide specific decoding parameters in Table 5. More details can be found in Appendix A.

**Baselines.** We compare our method against several mainstream data synthesis baselines. The simplest baseline is Self-Instruct (Wang et al., 2023), where we use the unoptimized teacher to synthesize data. Additionally, we select GPT-4o (OpenAI, 2024) as a stronger teacher to synthesize an equivalent amount of data for comparison. Another baseline is Self-Reward (Yuan et al., 2024), which employs an LLM-as-a-judge (Zheng et al., 2024) to assign ratings from 1 to 5 points to its self-synthesized responses. Since we find in our preliminary experiments that Llama3-8B lacks the ability to effectively score its own responses, we instead employ GPT-4o as an external judge to score the student's responses. The results of the original Self-Reward are reported in the Appendix § D.3. The final baseline is LLM2LLM (Lee et al., 2024), which evaluates the student's accuracy on its seed set and filters out those that result in incorrect answers. In our case, we define data points with the highest 50% training loss as incorrect examples. The teacher is then prompted to bootstrap data similar to the incorrectly answered seeds. To align with our setting, we uniformly conduct two rounds of iterations for Self-Reward and LLM2LLM. For all methods, we synthesize 10K instruction-response pairs to train the student models.

**Evaluation Methods.** We use Alpaca Eval 2.0 (Dubois et al., 2024) as the in-domain evaluation to assess the model's instruction-following ability. We utilize `gpt-4-turbo-2024-04-09` as the evaluator and uniformly compare all methods against the student model trained with Self-Instruct. The evaluation metrics are standard Winng Rate (WR) and Length Control Winning Rate (LC-WR). For head-to-head winning rate, we employ the evaluation prompt in both pairwise orders, and if the results disagree, we count it as a tie. Additionally, we evaluate the model's generalization performance across six out-of-domain tasks, including MT-Bench (Zheng et al., 2024), ARC-Challenge (25-shot) (Clark et al., 2018), GSM8K (8-shot) (Cobbe et al., 2021), HellaSwag (8-shot) (Zellers et al., 2019), GPQA (0-shot) (Rein et al., 2023), and MMLU (0-shot) (Hendrycks et al., 2020). These tasks span areas such as multi-turn dialogue, knowledge-based question answering, mathematics, and natural language reasoning, offering a thorough assessment of our approach's effectiveness. For MT-Bench, we report the score out of 10 judged by `gpt-4-turbo-2024-04-09`. For other tasks, we report normalized accuracy if it is included in the evaluation results, otherwise, standard accuracy.

## 5 EVALUATION RESULTS

This section evaluates the effectiveness of Montessori-Instruct (§ 5.1), illustrates the correlation between the teacher's learning and the student's performance (§ 5.2), conducts comprehensive ablation studies on the effectiveness of local data influence, the optimization of the teacher, the seed

Table 1: Evaluation of training 8B/1.1B students with different data synthesis methods. Adoption of a stronger teacher model (GPT-4o) is indicated by ∗. All else use Llama3-8B-Instruct as the teacher model. The best and second-best performances are marked in **bold** and underscore, respectively.

| Methods | In-Domain | | Out-Of-Domain | | | | | |
|---|---|---|---|---|---|---|---|---|
| | Alpaca Eval 2.0 | | MT-Bench | MMLU | GPQA | ARC-C | GSM8K | HellaSwag |
| | LC-WR | WR | Score | | | Accuracy | | |
| **8B Setting:** Student=Llama3-8B | | | | | | | | |
| *No fine-tuning* | 2.09% | 3.39% | 5.597 | 62.15 | 24.33 | 57.85 | 51.25 | 81.96 |
| *Self-Instruct* | 50% | 50% | 6.490 | 62.42 | **31.92** | 59.98 | 58.76 | 80.93 |
| *Self-Instruct** | 54.95% | 56.39% | 5.918 | 63.41 | 30.13 | 60.58 | 50.42 | 81.42 |
| *Self-Reward** | | | | | | | | |
| Iteration 1 | 51.87% | 55.38% | 6.713 | 62.46 | 28.19 | 59.84 | 53.60 | 81 .04 |
| Iteration 2 | 53.49% | 57.32% | 6.798 | 62.02 | 29.08 | 60.64 | 56.37 | 81.13 |
| *LLM2LLM* | | | | | | | | |
| Iteration 1 | 51.49% | 53.12% | 6.531 | 62.18 | 29.12 | 57.49 | 55.28 | 80.49 |
| Iteration 2 | 52.63% | 55.02% | 6.519 | 62.46 | 30.04 | 59.65 | 57.75 | 80.57 |
| *Montessori-Instruct* | | | | | | | | |
| Iteration 1 | 54.92% | 58.59% | 6.903 | 62.93 | 29.91 | 62.97 | 58.76 | 81.22 |
| Iteration 2 | **56.37%** | **60.15%** | **7.163** | **63.47** | 31.36 | 60.17 | **60.02** | **81.98** |
| **1.1B Setting:** Student=Tinyllama-1.1B | | | | | | | | |
| *No fine-tuning* | 17.89% | 17.56% | 1.020 | 26.16 | 23.88 | 37.12 | 1.97 | 62.61 |
| *Self-Instruct* | 50% | 50% | 2.154 | 26.21 | 24.78 | 37.97 | 1.82 | 62.47 |
| *Self-Instruct** | 54.02% | **55.02%** | 1.928 | **26.64** | 24.33 | **38.82** | 2.20 | 63.17 |
| *Self-Reward** | | | | | | | | |
| Iteration 1 | 47.62% | 48.34% | 1.804 | 26.34 | 23.92 | 37.64 | 1.76 | 62.27 |
| Iteration 2 | 46.48% | 46.95% | 1.717 | 26.09 | 24.62 | 38.03 | 1.76 | 62.79 |
| *LLM2LLM* | | | | | | | | |
| Iteration 1 | 52.03% | 52.75% | 2.243 | 25.87 | 24.51 | 36.86 | 2.24 | 62.15 |
| Iteration 2 | 51.64% | 53.52% | 2.192 | 25.62 | 24.84 | 36.74 | 2.31 | 62.08 |
| *Montessori-Instruct* | | | | | | | | |
| Iteration 1 | 53.25% | 51.77% | 2.485 | 26.23 | 23.92 | 37.97 | 2.35 | 62.59 |
| Iteration 2 | **54.37%** | 54.68% | **2.526** | 26.47 | 24.88 | 38.05 | **2.82** | **63.54** |

data and multiple iterations (§ 5.3), and then demonstrates the generalization of the synthetic data from Montessori-Instruct (§ 5.4).

## 5.1 OVERALL PERFORMANCE

Table 1 presents the overall performance of Montessori-Instruct compared with the state-of-the-art data synthesis methods. In the 8B setting, Montessori-Instruct significantly outperforms Self-Instruct by 6.37% LC-WR and 10.15% WR on Alpaca Eval. Notably, our method still surpasses Self-Instruct with GPT-4o as the teacher, suggesting that a stronger LLM does not necessarily produce more beneficial data than a weaker LLM that is tailored to the student's needs. Compared to Self-Reward and LLM2LLM, Montessori-Instruct consistently shows better performance across both iterations. This underscores the advantage of directly optimizing the teacher model's parameters toward the student's preferences derived from data influence.

In addition to in-domain evaluation, Montessori-Instruct also outperforms all the baselines on out-of-domain tasks, achieving maximum improvements of 0.673 and 0.372 on the MT-Bench in the 8B and 1.1B settings, respectively. This indicates that the teacher optimized by our method does not overfit the reference tasks and maintains strong robustness and generalization capabilities, whereas other baselines suffer from performance degradation on out-of-domain tasks.

## 5.2 CORRELATION BETWEEN TEACHER'S LEARNING AND STUDENT'S PERFORMANCE

This set of experiments examines how the teacher is progressively optimized to align with student preferences, thereby enhancing the student's performance. We first zoom in on the teacher's learning process to investigate its progressive impact on student models. Figures 3a and 3b compare the

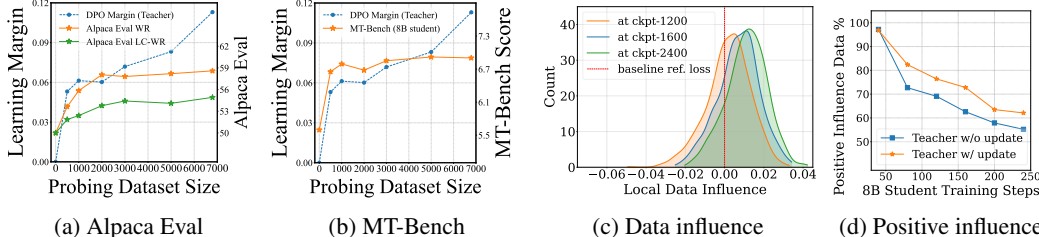

Figure 3: Figures (a) and (b) illustrate the correlation between the teacher's learning process and the performance of the student trained on data synthesized by the intermediate teachers in Alpaca Eval and MT-Bench. Figure (c) depicts how the distribution of the local data influence of the teacher's synthetic data shifts as the teacher is progressively updated. Figure (d) presents the proportion of training data with positive local data influence during the student's training.

performance of students trained using synthetic data generated from the teacher's intermediate checkpoints. The learning margin reflects the teacher's learning process, representing the average difference between selected rewards and corresponding rejected rewards in DPO. A larger margin indicates that the teacher is more likely to generate the selected synthetic data. The results indicate a positive correlation between the student's performance and the teacher's optimization progress.

We then select several teacher checkpoints to examine the properties of their synthetic data, aiming to identify changes occurring as the teacher learns. Specifically, we focus on the distribution of local data influence in the synthetic data, defined as the change in the model's reference loss before and after training on a single data point, which indicates the utility of that data for the model. The baseline reference loss is the loss on the reference set prior to one-step training, i.e., Equation 2. As shown in Figure 3c, we observe that as the teacher is optimized, the distribution of its synthetic data shifts towards the positive side, indicating an increased proportion of data with positive local influence in its synthetic outputs. From the student's perspective (Figure 3d), which shows the changes in the proportion of data with positive local influence in the next training batch, this proportion decreases over time during training. However, the data generated by the updated teacher consistently maintains a higher proportion of positive influence compared to a regular teacher.

In summary, we attribute the improved performance achieved by Montessori-Instruct to the teacher's continuously enhanced ability to synthesize data with higher local influence, by using DPO to distinguish data with varying influence values. The positive correlation between student performance and the increased proportion of training data with positive local influence leads to more effective learning, thereby improving the student's overall performance.

## 5.3 ABLATION STUDIES

This subsection demonstrates the effectiveness of the methodological design in Montessori-Instruct through four ablation studies, summarized in Table 2. The yellow lines show ablations on data point utility evaluation methods. The red lines represent optimization for **responses** based on instructions and optimization for **teacher models**. The blue lines cover various seed data types: OOD (**O**ut-**O**f-**D**omain), ID (**I**n-**D**omain), and Test (direct use of the **test set**).

**Effectiveness of Local Data Influence.** To evaluate the impact of different methods for obtaining the influence of a data point, we compare our local data influence against two additional baselines: (1) LLM-as-a-Judge (Zheng et al., 2024), which leverages GPT-4o to directly assign a 1-5 score to each instruction-response pair, inspired by Self-Reward, and (2) Training loss, which directly uses the training loss of each data point as its influence score, inspired by LLM2LLM. As shown in the yellow lines in table 2, our local data influence consistently outperforms both baselines by a significant margin. This indicates that local data influence is a more effective metric for capturing students' fine-grained data preferences compared to the other methods.

**Effectiveness of Teacher Optimization.** To analyze the effectiveness of the optimization strategy on the teacher, we compare our method with two additional ablation baselines: (1) Bootstrap: we

Table 2: Ablation studies on the effectiveness of the methodological design in Montessori-Instruct. All experiments were conducted on the Llama3-8B students.

| Methodological design | Alpaca Eval 2.0 | | MT-Bench | MMLU | GPQA | ARC-C | GSM8K | HellaSwag |
|---|---|---|---|---|---|---|---|---|
| | LC-WR | WR | Score | | | Accuracy | | |
| **Effectiveness of Local Data Influence** | | | | | | | | |
| LLM-as-a-Judge | 53.42% | 54.93% | 6.731 | **62.93** | 29.75 | 62.09 | **58.82** | 81.05 |
| Training loss | 52.34% | 54.99% | 6.656 | 62.54 | 29.89 | 61.48 | 58.76 | 80.93 |
| Local data influence (Ours) | **54.92%** | **58.59%** | **6.903** | **62.93** | **29.91** | **62.97** | 58.76 | **81.22** |
| **Effectiveness of Teacher Optimization** | | | | | | | | |
| Bootstrap | 50.59% | 48.14% | 6.618 | 60.67 | 25.19 | 57.95 | 58.13 | 80.46 |
| Response optimization | 51.59% | 54.22% | 6.556 | 62.43 | 27.45 | 60.42 | 56.38 | 81.04 |
| Instruction optimization (Ours) | **54.92%** | **58.59%** | **6.903** | **62.93** | **29.91** | **62.97** | **58.76** | **81.22** |
| **Effectiveness of Seed Data** | | | | | | | | |
| Open Assistant (OOD) | 52.28% | 54.76% | 6.706 | 62.86 | 29.74 | 62.29 | 58.42 | **81.24** |
| Alpaca GPT4 (ID) (Ours) | 54.92% | 58.59% | 6.903 | **62.93** | 29.91 | 62.97 | 58.76 | 81.22 |
| Alpaca Eval (Test) | **57.64%** | **61.36%** | **7.147** | **62.93** | **30.44** | **63.06** | **60.80** | 81.09 |

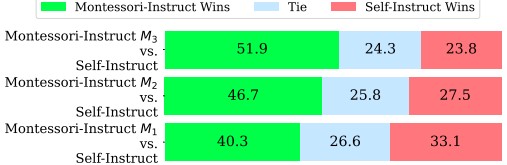

(a) Win rates of iterations compared to Self-Instruct

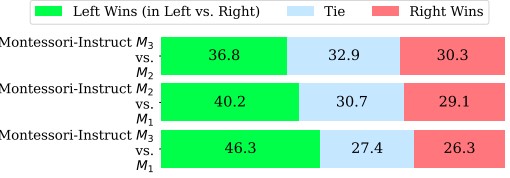

(b) Win rates compared between different iterations

Figure 4: Head-to-head win rates for evaluating 8B models among the Self-Instruct baseline and three successive iterations updated using Montessori-Instruct.

bootstrap the top 50% influential data by utilizing it as the seed, and (2) Response optimization: we optimize the teacher by the student's local data influence of different responses given an instruction. As shown in red lines in table 2, optimizing the teacher is generally better than merely bootstrapping influential data, highlighting the necessity of adapting the teacher to the student's needs. Furthermore, instruction optimization (Montessori-Instruct) outperforms response optimization across all tasks. We attribute this to the smaller search space of response optimization, which limits the headroom for teacher improvement compared to instruction optimization.

**Effectiveness of Seed Data.** This study examines the impact of the seed data by varying its relevance to the evaluation tasks. In addition to the Alpaca GPT-4 (in-domain seed data) used in the main experiments, we also utilize Open Assistant and Alpaca Eval as alternative seed data. Open Assistant represents an out-of-domain seed, whereas Alpaca Eval is directly sampled from the evaluation task. Blue lines in table 2 demonstrates that using Alpaca Eval leads to the best performance on itself while using Open Assistant is less effective compared to in-domain seed data. For more general NLP benchmarks, changing the seed data results in only slight differences in performance. This indicates that our method is robust enough to enhance the synthesis ability of teachers, even when using different seeds.

**Effectiveness of Multiple Iterations.** We examine the performance differences when applying Montessori-Instruct over multiple iterations. In each iteration, we begin by constructing a probing dataset of 2K samples to collect local data influence on the student model from the previous iteration, followed by updating the previous teacher. As shown in Figure 4a, Montessori-Instruct continues to outperform Self-Instruct across three iterations, achieving a peak head-to-head win rates of 51.9%. The results in Figure 4 illustrate the comparison between different iterations, demonstrating that Montessori-Instruct can yield improvements over previous iterations. We attribute these gains to the Montessori-Instruct's ability to capture the data preferences of students at different iterations and to tailor influential data according to their evolving needs.

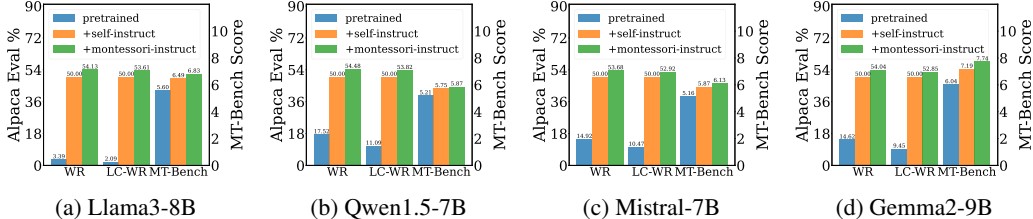

(a) Llama3-8B     (b) Qwen1.5-7B     (c) Mistral-7B     (d) Gemma2-9B

Figure 5: Evaluation results of training four different student models using synthetic data generated by a teacher optimized for the data preferences of the 1.1B student.

## 5.4 GENERALIZATION ABILITY OF THE SYNTHESIZED DATA

In this experiment, we study the generalization ability of our teacher optimized toward a small student (1.1B)'s preferences. Specifically, we utilize the data synthesized by this teacher to train four different student models—Llama3-8B (Meta, 2024), Mistral-7B (Jiang et al., 2023), Qwen1.5-7B (Bai et al., 2023), and Gemma2-9B (Team et al., 2024). As shown in Figure 5, the data synthesized by one teacher leads to consistent performance gains across all the students compared to Self-Instruct. This finding implies we can directly deploy an optimized teacher to generate data for a variety of student models, enhancing their performance with a low expense.

## 5.5 CASE STUDY

In this section, we present several cases to visualize the differences between the instructions synthesized by Self-Instruct and by Montessori-Instruct, and showcase the chosen and rejected data pairs that reflect what the teacher learns during our optimization. Figure 6 shows the word analysis of root verbs and their corresponding nouns. We identify the top 10 most common root verbs (inner circle) and their top 4 direct noun objects (outer circle) in the generated instructions. The results indicate that, compared to Self-Instruct, Montessori-Instruct guides the teacher to synthesize more

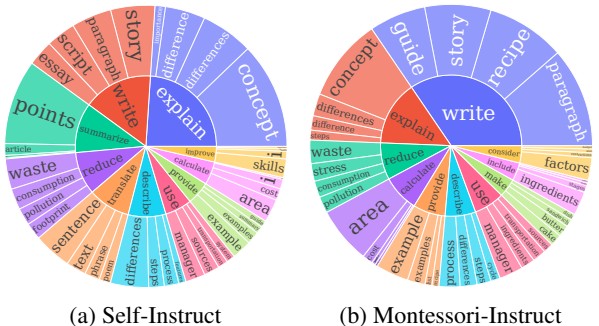

(a) Self-Instruct     (b) Montessori-Instruct

Figure 6: The most common root verbs (inner circle) and their top direct noun objects (outer circle) in generated instructions

on writing instructions and providing specific, informative examples, while reducing the frequency of simple commands like summarizing and translating.

Table 3 compares the chosen and rejected data pairs given the same prompt. Our method discards low-utility data, such as explanations of simple concepts and sentence translations, and increases the likelihood of generating complex and informative instructions. This further demonstrates the effectiveness of using local data influence to differentiate data utility.

## 6 DISCUSSION AND LIMITATIONS

**Synthetic Data Scale.** We synthesize 10K data points to verify the effectiveness of our innovative data synthesis framework. While this 10K dataset outperforms other baselines and demonstrates strong generalization, its effectiveness when scaled to the volume required for production-level fine-tuning (around 100K) remains unclear. Expanding the synthetic data volume may introduce redundancy, a phenomenon commonly observed in data synthesis (Bauer et al., 2024; Liu et al., 2024). It would be meaningful to study how to balance the quantity and the diversity of the synthetic data, while this is orthogonal to our main contribution.

Table 3: Examples of instruction-response pairs with positive and negative influences

| Positive | Negative |
|---|---|
| **[Instruction]:** What is the most efficient way to transport a given package across a city, considering traffic congestion, distance, and cost?
**[Response]:** The most efficient way to transport a package across a city depends on several factors, including the package's size, weight, fragility, and urgency, as well as the city's traffic patterns, infrastructure, and regulations. Here are some general considerations and options to minimize costs, reduce congestion, and ensure timely delivery:
1. **Distance and Route Optimization**:
   * Use online route planners or transportation management software to determine the shortest and most efficient route.
   * Consider alternative routes or modes of transportation if the primary route is congested or closed.
2. **Traffic Congestion Mitigation**:
   * Avoid peak hours (7-9 am and 4-6 pm) when possible, as traffic congestion is typically highest during these times... | **[Instruction]:** Explain the concept of artificial intelligence in 100 words.
**[Response]:** Artificial intelligence (AI) refers to the development of computer systems that can perform tasks that typically require human intelligence, such as learning, problem-solving, and decision-making. AI systems use algorithms and data to analyze and interpret information, allowing them to make predictions, classify objects, and generate insights. They can be trained on large datasets to improve their performance over time, and can even adapt to new situations and learn from experience. AI has many applications, including robotics, natural language processing, image recognition, and expert systems, and is increasingly being used in industries such as healthcare, finance, and transportation. |
| **[Instruction]:** Can you write a script for a public service announcement about the importance of recycling?
**[Response]:** Here is a script for a public service announcement about the importance of recycling:
[Opening shot of a beautiful park or nature reserve, with a narrator speaking]
Narrator: "Imagine a world where our planet is thriving, where wildlife roams free, and where our communities are healthy and strong. But to get there, we need to take action."
[Cut to a shot of a person holding a plastic bottle and throwing it away]
Narrator: "One of the simplest and most effective ways to make a difference is by recycling."... | **[Instruction]:** Can you translate the following text from english to spanish: "the hotel room was comfortable and clean."
**[Response]:** The translation of the text from English to Spanish is:
"El cuarto de hotel era cómodo y limpio."
Here's a breakdown of the translation:
* "the hotel room" = "el cuarto de hotel"
* "was" = "era" (verb conjugation)
* "comfortable" = "cómodo"
* "and" = "y" (conjunction)
* "clean" = "limpio"
So, the complete sentence in Spanish is: "El cuarto de hotel era cómodo y limpio." |

**Overhead.** Montessori-Instruct introduces an additional computational cost. Compared to Wang et al. (2023), training an 8B model using our method increases the average processing time per data by 5.8 seconds (see the Appendix E for details). At the instruction finetuning stage, compute is less an issue compared to pretraining. The scale is smaller, and generating data is faster and cheaper than human annotations. Additionally, the most time-intensive step in our method–"collecting local data influence"–can be independently parallelized on heterogeneous compute systems, allowing for easy acceleration. As demonstrated in § 5.4, Montessori-Instruct exhibits strong generalization capabilities. In practice, one can use a smaller model to collect data influence for updating the teacher and then apply the updated teacher to synthesize data for larger models.

## 7 CONCLUSION

In this paper, we propose Montessori-Instruct, a novel data synthesis framework that tailors the teacher for student learning. Montessori-Instruct leverages local data influence to reflect the student's learning preferences and to optimize the teacher to produce more influential synthetic training data. Experimental results demonstrate that Montessori-Instruct significantly outperforms state-of-the-art data synthesis methods in both in-domain and out-of-domain evaluations, exceeding the performance of data generated by stronger teacher models like GPT-4o. Further analyses confirm the benefits of optimizing the teacher toward the student's preferences in improving student performances. Ablation studies validate the benefits of using local data influence to reflect data utility and highlight the benefits of optimizing the teacher over bootstrapping. Our work successfully demonstrates the potential of incorporating the student's learning preferences into teacher optimization, and we hope it inspires further exploration of more effective synthetic data generation frameworks.

ACKNOWLEDGEMENTS

We sincerely thank Shi Yu and Zhenghao Liu for discussing ideas and providing helpful feedback on this work. We also extend our gratitude to Alex Xu for testing our Github repository.

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

## A    TRAINING DETAILS

The hyperparameters used during training teachers and students are as follows. We employ the AdamW optimizer (Loshchilov & Hutter, 2019) with a WSD scheduler (Hu et al., 2024a). For SFT, the 8B model utilizes a maximum learning rate of $5e^{-6}$, while the 1B model uses $1e^{-5}$. The WSD scheduler is configured with a warmup ratio of $0.1$, a stable ratio of $0.5$, and a decay ratio of $0.4$, with the learning rate decaying to one-thousandth of the maximum. The epoch is set to $1$, batch size is set to $32$ and the dropout is $0$. We mask non-target tokens, calculating the loss only on target tokens. If the student model does not have a chat template itself, we apply the Llama3-8B formatted chat template, as shown in 7, with bos_token, eos_token and pad_token set to <|start_header_id|>, <|end_header_id|>, and <|end_header_id|>, respectively. For DPO, we use a learning rate of $1e^{-6}$, set $\beta$ to $0.1$, and use a batch size of $2$, while other parameters remain the same as in SFT.

Figure 7: Chat Template

```
{% if messages[0]['role'] == 'system' %}
    {% set offset = 1 %}
{% else %}
    {% set offset = 0 %}
{% endif %}

{{ bos_token }}
{% for message in messages %}
    {% if (message['role'] == 'user') !=
(loop.index0 % 2 == offset) %}
        {{ raise_exception('Conversation roles
must alternate userassistantuserassistant...')
}}
    {% endif %}

    {{ <|start_header_id|> + message['role'] +
<|end_header_id|> + message['content'] | trim +
eos_token }}
{% endfor %}

{% if add_generation_prompt %}
    {{ '<|start_header_id|>' + 'assistant' +
'<|end_header_id|>
n
n' }}
{% endif %}
```

We use Hugging Face TRL codebase (von Werra et al., 2020) to perform both full parameters fine-tuning and direct preference optimization. For the 8B model, we employ the Hugging Face Accelerate codebase (Gugger et al., 2022) to facilitate FSDP training (Zhao et al., 2023). All the parameters introduced in this section are summarized in Table 4.

Table 4: Training Parameters

| Method | Learning Rate | Weight Decay | Warmup Ratio | Stable Ratio | Decay Ratio |
|---|---|---|---|---|---|
| SFT | $5.0e-6$ | 0.0 | 0.1 | 0.5 | 0.4 |
| DPO | $1.0e-6$ | 0.0 | 0.1 | 0.5 | 0.4 |

| Method | Minium Learning Rate | Epoch | Per Device Train Batch Size | Gradient Accumulation | Train Batch Size |
|---|---|---|---|---|---|
| SFT | $5.0e-9$ | 1 | 2 | 2 | 32 |
| DPO | $1.0e-9$ | 1 | 2 | 1 | 2 |

| Method | Max Length | Dropout | BF16 | Flash Attention 2 | Beta |
|---|---|---|---|---|---|
| SFT | 1024 | 0.0 | True | True | - |
| DPO | 1024 | 0.0 | True | True | 0.1 |

## B  THEORETICAL GUARANTEE OF LOCAL DATA INFLUENCE

This section provides a detailed explanation of the derivation for computing local data influence and the rationale behind its effectiveness. We referred to the derivation method in Yu et al. (2024b). We use $\mathcal{D}_{\text{ref}}$ to represent the reference set and $m$ to represent the student model that we calculate local data influence on. The derivation begins with the standard influence functions Koh & Liang (2017); Weisberg & Cook (1982) which quantify the change in reference loss when a data point $x_i$ is upweighted by a small $\epsilon$. We denote the optimal model state after the upweighting as $m_{\epsilon, x_i} = \arg\min_m \frac{1}{n} \sum_{j=1}^n \mathcal{L}(x_j \mid m) + \epsilon \mathcal{L}(x_i \mid m)$ and simplify the optimal model under $\epsilon = 0$ case (i.e., no upweighting) as $m$. The influence of upweighting $x_i$ is then given by:

$$\mathcal{I}_m(x_i; \mathcal{D}_{\text{ref}}) \overset{\text{def}}{=} \frac{d\mathcal{L}(\mathcal{D}_{\text{ref}} \mid m_{\epsilon, x_i})}{d\epsilon}\Big|_{\epsilon=0} \tag{5}$$

$$= \nabla_m \mathcal{L}(\mathcal{D}_{\text{ref}} \mid m)^\top \frac{dm_{\epsilon, x_i}}{d\epsilon}\Big|_{\epsilon=0} \tag{6}$$

$$= -\nabla_m \mathcal{L}(\mathcal{D}_{\text{ref}} \mid m)^\top H_m^{-1} \nabla_m \mathcal{L}(x_i \mid m), \tag{7}$$

where $H_m = \frac{1}{n} \sum_{j=1}^n \nabla_m^2 \mathcal{L}(x_j \mid m)$ is the Hessian matrix, which is positive definite. The derivation from Eq. 6 to Eq. 7 is given by building a quadratic approximation to the empirical risk around $m$ and tperforming a single Newton step as shown in Koh & Liang (2017). Now let's consider the scenario in which $x_i$ is incorporated into the training data. In this case, $\epsilon = \frac{1}{n}$, and the parameter difference due to the inclusion of $x_i$ is $m_{\frac{1}{n}, x_i} - m \approx \frac{1}{n} H_m^{-1} \nabla_m \mathcal{L}(x_i \mid m)$ and the influence in Eq. 7 can be further represented as:

$$\mathcal{I}_m(x_i; \mathcal{D}_{\text{ref}}) \approx n \nabla_m \mathcal{L}(\mathcal{D}_{\text{ref}} \mid m)^\top (m_{\frac{1}{n}, x_i} - m) \tag{8}$$

$$\approx n(\mathcal{L}(\mathcal{D}_{\text{ref}} \mid m_{\frac{1}{n}, x_i}) - \mathcal{L}(\mathcal{D}_{\text{ref}} \mid m)) \tag{9}$$

$$\propto -\mathcal{L}(\mathcal{D}_{\text{ref}} \mid m) + \mathcal{L}(\mathcal{D}_{\text{ref}} \mid m_{\frac{1}{n}, x_i}). \tag{10}$$

So far, we have successfully derived the method (Eq. 10) of calculating local data influence used in § 3.2. Using the supervised fine-tuning algorithm $\mathcal{A}$, we denote the model state $m_{\frac{1}{n}, x_i}$ as $\mathcal{A}(y_i \mid x_i; m)$, which is updated on the synthetic data point $(x_i, y_i)$ for one step. Replacing the variables in Eq. 10 with the notation of our method, we can obtain:

$$\mathcal{I}_m(x_i; \mathcal{D}_{\text{ref}}) \approx -\mathcal{L}(\mathcal{D}_{\text{ref}} \mid \mathcal{A}(y_i \mid x_i; m)) + \mathcal{L}(\mathcal{D}_{\text{ref}} \mid m) \tag{11}$$

## C  STATISTICS ON SYNTHESIS DATA

We plot the top 20 most common root verbs (inner circle) and their top 4 direct noun objects (outer circle) in the generated instructions by Self-Instruct (Figure 8), the first iteration of Montessori-Instruct (Figure 9), and the second iteration of Montessori-Instruct (Figure 10), respectively.

We observe an increasing trend in instructions such as 'write,' 'provide,' and 'make,' as well as a consistent trend for instructions like 'explain' and 'describe.' These commands typically require more general detailed information and lead to longer, more complex responses. Meanwhile, commands like 'translate' and 'calculate' show a decline, as they usually require straightforward answers and simpler formats. This outcome demonstrates that Montessori-Instruct helps the teacher model generate more detailed and informative instructions, thereby improving student performance.

We also plot the distribution of tokenized instructions and responses generated by Self-Instruct and Montessori-Instruct for comparison. As shown in Figures 11 and 12, there is an increasing trend in the length of instructions, while the length of responses remains relatively unchanged. This aligns with our design, which focuses on optimizing instructions based on prompts rather than optimizing responses based on instructions. The increased length of instructions also reflects the teacher's data synthesis strategy shifting toward more complex and informative instructions.

Figure 8: The top 20 most common root verbs (inner circle) and their top 4 direct noun objects (outer circle) in the generated instructions by Self-Instruct

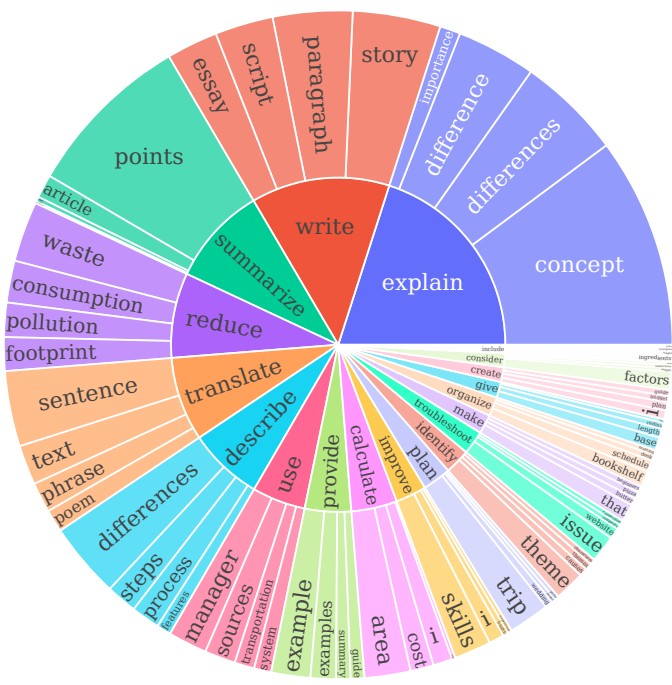

Figure 9: The top 20 most common root verbs (inner circle) and their top 4 direct noun objects (outer circle) in the generated instructions by Montessori-Instruct (iteration 1)

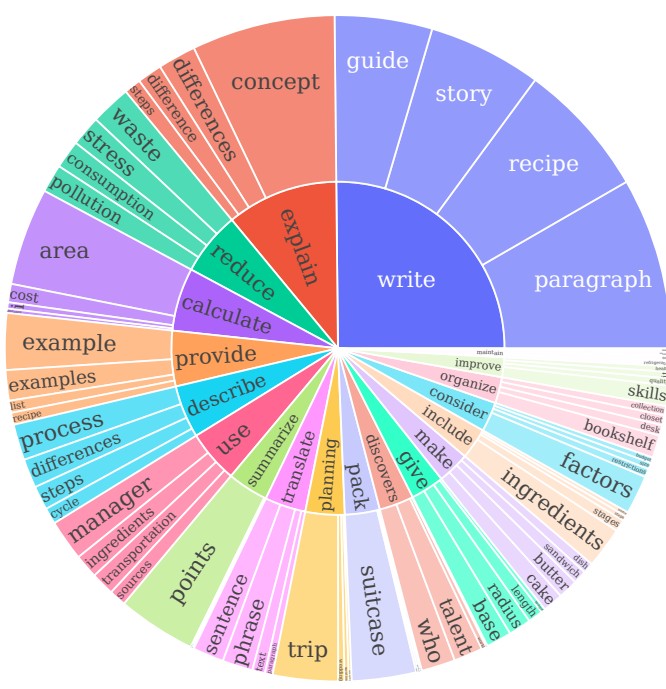

Figure 10: The top 20 most common root verbs (inner circle) and their top 4 direct noun objects (outer circle) in the generated instructions by Montessori-Instruct (iteration 2)

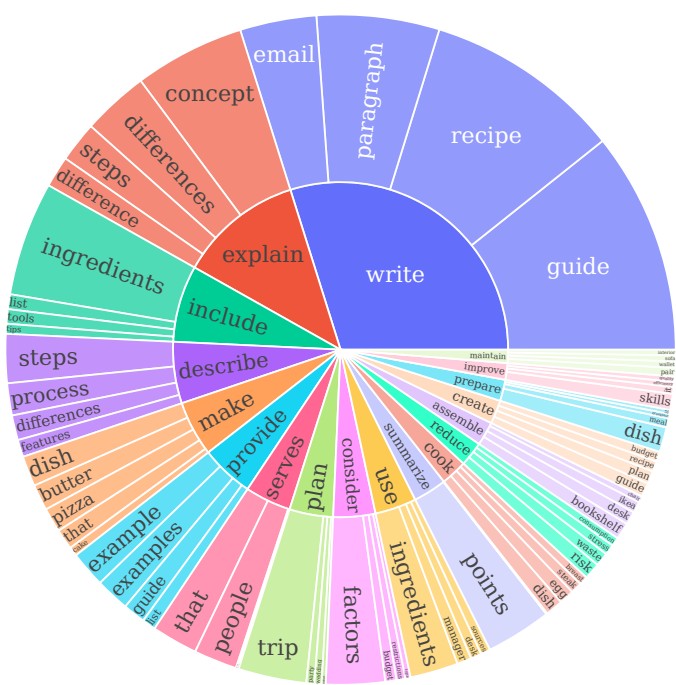

Figure 11: Distribution of tokenized **instructions** generated by Self-Instruct and Montessori-Instruct

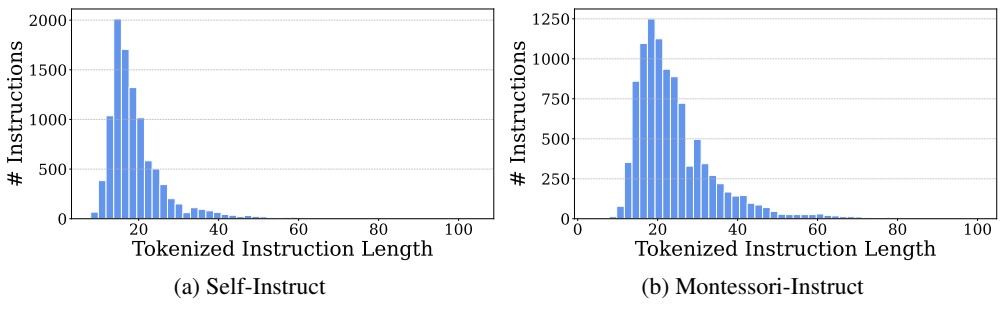

(a) Self-Instruct

(b) Montessori-Instruct

Figure 12: Distribution of tokenized **responses** generated by Self-Instruct and Montessori-Instruct

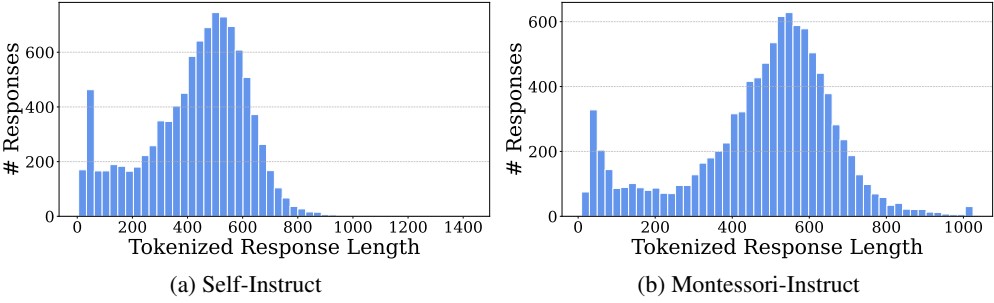

(a) Self-Instruct

(b) Montessori-Instruct

## D ADDITIONAL EXPERIMENTAL DETAILS

### D.1 PROMPTS USED FOR INSTRUCTION GENERATION.

In this section, we present the prompts used in Montessori-Instruct. Figure 13 illustrates how we prompt the teacher model to generate new instructions. We begin by outlining some requirements for the teacher, followed by inserting 8-shot seed examples sampled from both the seed pool and the data pool generated in the previous iteration. We then extract the instruction from the teacher's output using regex matching and filter out those with incorrect formats.

Figure 14 displays the prompt used in our ablation studies on the effectiveness of Local Data Influence. In this study, we evaluated different methods for assessing the utility of synthetic data, one of which involved using LLM-as-a-Judge (Zheng et al., 2024). We adapted the prompt from Self-Reward (Yuan et al., 2024) and added an additional point to evaluate the quality of the instruction, resulting in a maximum score of 6 points.

Figure 13: Prompt for Generating Instructions

> **Prompt**
>
> Generate an instruction. This instruction should be a question that humans would be ask. It can be in imperative or interrogative form. We will use the instructions you generate to train models, so you must ensure that the instructions generated are of high quality and correct and also keep the instruction clear and concise.
> You should:
> 1. Briefly explain why you generate this instruction.
> 2. Think about whether you need to add some input to this instruction so that it can be answered directly. (For example, for tasks that involve summarizing, you need to provide the paragraph to be summarized).
> 3. Return you output strictly following the format:
> Your generated instruction should strictly follow the following format:
> `<instruction><YOUR INSTRUCTION HERE><YOUR INPUT HERE></instruction>`
> If there is no need to add inputs to answer the instruction, you can skip the `<YOUR INPUT HERE>` part. If you need to add inputs, just replace the `<YOUR INPUT HERE>` with the input. Now here are some examples of reference instructions, and please generate only one instruction.

### D.2 DECODING STRATEGIES

We list all the parameters used for decoding outputs from language models in Table 5. Separate parameters are used for generating instructions and responses. A higher temperature is used for instruction generation to encourage diversity, enabling us to leverage local data influence to identify more informative instructions. For responses, we use a temperature of 0.6 to reduce uncertainty. Additionally, two penalty techniques are employed to mitigate duplication issues during synthesis.

### D.3 SELF-REWARD RESULTS WITHOUT THE EXTERNAL JUDGE

In this section, we report the results of the original Self-Reward (Yuan et al., 2024) method. Self-Reward requires the student model to generate responses to given instructions, and then assess their own responses by generating judgments and scores ranging from 1 to 5 using LLM-as-a-Judge (Zheng et al., 2024). It then employs Direct Preference Optimization (DPO) to encourage the student to synthesize higher-scoring responses. However, this approach demands a high level of instruction-following ability from the student model. The authors of Self-Reward employ Llama2-70B as the

Figure 14: LLM-as-a-Judge Prompt for evaluating instructions and corresponding responses in our ablation studies on the effectiveness of Local Data Influence

---

**Prompt**

Review the user's instruction and the corresponding response using the additive 6-point scoring system described below. Points are accumulated based on the satisfaction of each criterion:
- Add 1 point if the response is relevant and provides some information related to the user's inquiry, even if it is incomplete or contains some irrelevant content.
- Add another point if the response addresses a substantial portion of the user's question, but does not completely resolve the query or provide a direct answer.
- Award a third point if the response answers the basic elements of the user's question in a useful way, regardless of whether it seems to have been written by an AI Assistant or if it has elements typically found in blogs or search results.
- Grant a fourth point if the response is clearly written from an AI Assistant's perspective, addressing the user's question directly and comprehensively, and is well-organized and helpful, even if there is slight room for improvement in clarity, conciseness or focus.
- Bestow a fifth point for a response that is impeccably tailored to the user's question by an AI Assistant, without extraneous information, reflecting expert knowledge, and demonstrating a high-quality, engaging, and insightful answer.
- Award an additional point if you consider this instruction to be of moderate difficulty, requiring thought and analysis rather than being a straightforward task.
User:
`<INSTRUCTION_HERE>`
`<response><RESPONSE_HERE></response>`
After examining the user's instruction and the response:
- Briefly justify your total score, up to 100 words.
- Conclude with the score using the format: `\Score:`
`<total points>`"
Remember to assess from the AI Assistant perspective, utilizing web search knowledge as necessary. To evaluate the response in alignment with this additive scoring model, we'll systematically attribute points based on the outlined criteria.

---

Table 5: Decoding Parameters using vLLM

|  | Generate Instruction | Generate Responses |
|---|---|---|
| temperature | 1 | 0.6 |
| top_p | 0.9 | 0.9 |
| frequency_penalty | 0 | 0 |
| presence_penalty | 1 | 1 |
| repetition_penalty | 1.5 | 1 |
| max_token | 1024 | 1024 |

student model for this reason. In our experimental setup with Llama3-8B and TinyLlama-1.1B, both models lack sufficient instruction-following capabilities and fail to produce detailed judgments and valid scores. For example, Llama3-8B's scores are skewed, clustering around 4 and 5, making it difficult to differentiate between responses. The 1.1B model's scores even do not follow the rules in

Table 6: Evaluation of training 8B/1.1B students using the original Self-Reward settings compared to Self-Instruct, without relying on external judges.

| Methods | In-Domain | | Out-Of-Domain | | | | | |
|---|---|---|---|---|---|---|---|---|
| | Alpaca Eval 2.0 | | MT-Bench | MMLU | GPQA | ARC-C | GSM8K | HellaSwag |
| | LC-WR | WR | Score | | | Accuracy | | |
| **8B Setting:** Student=Llama3-8B | | | | | | | | |
| *No fine-tuning* | 2.09% | 3.39% | 5.597 | 62.15 | 24.33 | 57.85 | 51.25 | 81.96 |
| *Self-Instruct* | 50% | 50% | 6.490 | 62.42 | 31.92 | 59.98 | 58.76 | 80.93 |
| *Self-Reward* | | | | | | | | |
| Iteration 1 | 2.45% | 4.06% | 5.442 | 61.79 | 24.30 | 57.81 | 49.92 | 80.75 |
| Iteration 2 | 2.69% | 4.71% | 5.428 | 61.79 | 23.58 | 57.64 | 49.53 | 80.17 |
| **1.1B Setting:** Student=Tinyllama-1.1B | | | | | | | | |
| *No fine-tuning* | 17.89% | 17.56% | 1.020 | 26.16 | 23.88 | 37.12 | 1.97 | 62.61 |
| *Self-Instruct* | 50% | 50% | 2.154 | 26.21 | 24.78 | 37.97 | 1.82 | 62.47 |
| *Self-Reward* | | | | | | | | |
| Iteration 1 | 7.79% | 8.13% | 1.000 | 23.58 | 22.30 | 36.55 | 0.94 | 61.92 |
| Iteration 2 | 6.34% | 7.57% | 1.000 | 23.44 | 22.06 | 36.49 | 0.98 | 61.24 |

the prompt and fall outside the specified 1 to 5 range. Therefore, in our main experiment, we use GPT-4o as an external judge to score the student responses. Nonetheless, we also report results here based on the original Self-Reward settings, where the model judges its own responses without relying on a more powerful external model.

# E   COST ANALYSIS

## E.1   TIME OVERLOAD

Compared to Self-Instruct (Wang et al., 2023), our method introduces additional overhead in: (1) collecting local data influence to construct the preference dataset (§ 3.2), (2) and performing DPO optimization for the teacher model (§ 3.3). The majority of the computational overhead arises from collecting local data influence. This process begins by generating instructions and responses to create a probing dataset, distinct from the training set used for fine-tuning the student, and used solely for calculating local data influence. Then, we traverse the entire probing dataset, fine-tuning the student model on each individual data point to collect its corresponding local influence. For each data point, loading the student's warmed-up checkpoint from disk, training for one step, and evaluating on the reference dataset are the primary time-consuming steps. We provide a detailed breakdown of the time required for these steps in table 7 and calculate the average time needed to run the entire Montessori-Instruct process and resulte in the final student model. The calculations are based on a probing dataset and training dataset, each consisting of 10K entries.

However, there are two simple ways to reduce the time demand for Montessori-Instruct. First, the process of collecting local data influence can be parallelized independently on a heterogeneous compute system to speed up execution, with no need for communication between systems—a common bottleneck in distributed training. In our experiments, we utilize 8 H100 GPUs to accelerate this process. Second, as demonstrated in our experiments (§ 5.4), Montessori-Instruct shows strong generalization capabilities. In practice, a smaller model can be used to collect data influence for updating the teacher, which can then synthesize data for larger models. This approach significantly reduces the computational overhead compared to using larger models directly for collecting local data influence.

Table 7: Time Overload Statistics

| Task | Sub task | 8B | 1B |
|---|---|---|---|
| collect local data influence / per data | generate instructions | 0.372s | |
| | generate responses | 0.031s | |
| | load warmuped ckpt from disk | 2.69s | 1.08s |
| | fine-tune for one step | 4.12s | 0.79s |
| | eval on reference set | 4.19s | 1.26s |
| | total | 13.403s | 3.533s |
| **Task** | | **8B** | **1B** |
| Time for DPO Training / per data | | 0.362s | |
| **Task** | **Method** | **8B** | **1B** |
| Time for obtaining the final student model / per data | Self-Instruct | 0.486s | 0.422s |
| | Montessori-Instruct | **5.842s** | **1.834s** |

### E.2 COST-PERFORMANCE RELATIONSHIP

We provide further clarification on the cost-performance relationship of our method compared to all baselines. We analyzed the Performance-FLOPs curve of four methods, with a particular focus on the changes in Self-Instruct's Alpaca Eval and MT-Bench Score as their FLOPs increase to levels comparable to those of Montessori-Instruct. We scale the FLOPs of Self-Instruct by synthesizing additional data. We also marked the Performance-FLOPs relationship of the two baselines, LLM2LLM and Self-Reward, in the following figures.

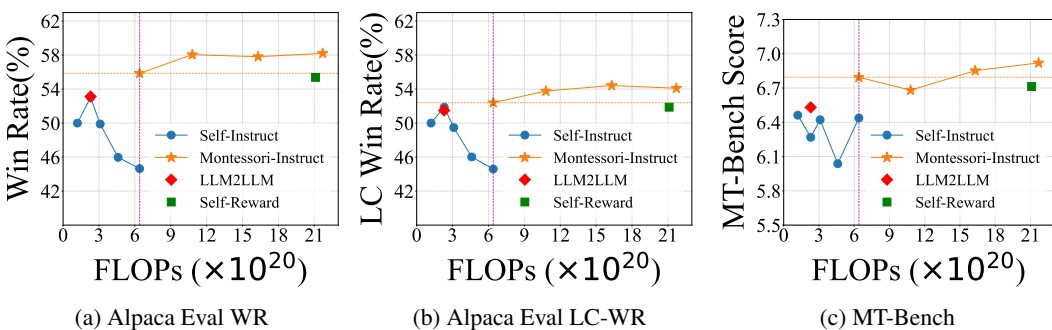

(a) Alpaca Eval WR      (b) Alpaca Eval LC-WR      (c) MT-Bench

Figure 15: The Performance-FLOPs curve for all four methods.

It can be seen that Self-Instruct quickly reached the upper bound during the scaling-up process, and even with more FLOPs, no better performance improvement can be achieved. The reason may be that the data generated by Self-Instruct is severely homogenized. In contrast, the upper bound of our method is significantly better and continuously grows when we invest more FLOPs into it.

Then we give a computational result of the FLOPs estimated for four methods, as well as the pretraining and test-time-scaling. The detailed derivation is provided in E.3. The main FLOPs for Montessori-Instruct come from processing probing data. In the Table 1, we used 10K probing data to utilize the most resources to achieve the best performance, but as the Figure 3a and Figure 3b suggests, using around 1K probing data can already achieve better performance than other baselines. To make a fair comparison, we calculate the FLOPs under 1K probing data. We estimate the FLOPs as follows (Llama3-8B-Instruct as the teacher, Llama3-8B as the student):

- Self-Instruct: $1.34 \times 10^{20}$ FLOPs
- Self-Reward: $2.11 \times 10^{21}$ FLOPs
- LLM2LLM: $2.3 \times 10^{20}$ FLOPs
- Montessori-Instruct: $6.43 \times 10^{20}$ FLOPs

- Pretrain Llama3-8B: $1.87 \times 10^{24}$ FLOPs
- Inference-Time Scaling: $1.60 \times 10^{23}$ FLOPs

We can see that Montessori-Instruct's FLOPs are 7 times less than Self-Reward. Furthermore, if we use the proxy model (Yu et al., 2024b), such as a smaller-sized model (e.g., 1B parameters for assisting an 8B model) to process probing data, Montessori's FLOPs can further reduce to $1.92 \times 10^{20}$ FLOPs. This makes it comparable to Self-Instruct while still outperforming it. Using a proxy model has promising potential for enhancing both efficiency and performance, which we leave for future work. Regarding the pretraining, since the computational cost during the SFT phase is significantly lower than that during the pretraining phase ($10^4$ times smaller), even if we increase resource investment in SFT, its overall consumption remains minimal. Recent work has focused on scaling inference time to achieve better performance (Snell et al., 2024). However, the inference-time scaling FLOPs are also significantly larger than those of SFT, being approximately $10^3$ times greater, according to Sardana et al. (2023). Nevertheless, our teacher training represents a one-time cost. As demonstrated in Section 5.4, the optimized teacher can assist multiple students in improving their performance without the need for retraining from scratch.

### E.3 DERIVATION OF FLOPS

- When generating synthetic data, the input window includes both **prompt** and **seed data**, so we set the **input length** to 2048.
- For **instruction-based input/output**, the **input/output length** is 128.
- For **response-based input/output**, the **input/output length** is 1024.
- For **judgment-based input/output** using an LLM, the **input/output length** is 1024.

We define the computational cost of generating one token for an input of length 128 as one unit $\mathcal{F}$. During instruction fine-tuning, the input and output lengths are 128 and 1024, respectively. The backward FLOPs are approximately twice the forward FLOPs. For one data sample, the training FLOPs can be estimated as:

$$1024\mathcal{F} \times 3 = 3072\mathcal{F}$$

FLOPs calculations are based on calflops (2024), where $\mathcal{F} = 1.92T$ FLOPs.

| Method | FLOPs ($\mathcal{F}$) |
|---|---|
| **Self-Reward** | |
| Synthesize 10K instructions from seed | $16\mathcal{F} \times 128 \times 10\text{K} = 20480\text{K}\mathcal{F}$ |
| Synthesize 4 responses per instruction | $40\text{K} \times 1024\mathcal{F} = 40960\text{K}\mathcal{F}$ |
| Generate 3 judgments per response | $40\text{K} \times 8\mathcal{F} \times 1024 \times 3 = 983040\text{K}\mathcal{F}$ |
| Train with 10K pairs using DPO | $\text{DPO}_{10\text{K}}$ |
| Synthesizes $2K$ instruction-response-judge sets | $(16\mathcal{F} \times 128 + \mathcal{F} \times 1024 + 8\mathcal{F} \times 1024)$ $\times 2\text{K} = 22528\text{K}\mathcal{F}$ |
| Perform SFT on student | $\text{SFT}_{2\text{K}}$ |
| **Total** | $\approx 1100\text{M}\mathcal{F} + \text{DPO}_{10\text{K}}$ |
| **LLM2LLM** | |
| Synthesize 10K instructions from seed | $16\mathcal{F} \times 128 \times 10\text{K} = 20480\text{K}\mathcal{F}$ |
| Generate 1 response per instruction | $10240\text{K}\mathcal{F}$ |
| Student responds to each instruction | $\mathcal{F} \times 1024 \times 10\text{K} = 10240\text{K}\mathcal{F}$ |
| Resynthesize 10K instructions | $20480\text{K}\mathcal{F}$ |
| Generate 1 response per instruction | $10240\text{K}\mathcal{F}$ |
| Perform SFT on student | $\text{SFT}_{10\text{K}}$ |
| **Total** | $\approx 120\text{M}\mathcal{F}$ |
| **Montessori** | |
| Synthesize 1K instructions from seed | $2048\text{K}\mathcal{F}$ |
| Generate 1 response per instruction | $1024\text{K}\mathcal{F}$ |
| Train student with each instruction | $\text{SFT}_{10\text{K}}$ |
| Evaluate trained student on validation set | $1\text{K}\mathcal{F} \times 1024 \times 256 = 262144\text{K}\mathcal{F}$ |
| Perform DPO updates on teacher with 6K samples | $\text{DPO}_{1\text{K}}$ |
| Resynthesize 10K instructions | $20480\text{K}\mathcal{F}$ |
| Generate 1 response per instruction | $10240\text{K}\mathcal{F}$ |
| Perform SFT on student | $\text{SFT}_{10\text{K}}$ |
| **Total** | $\approx 340\text{M}\mathcal{F} + \text{DPO}_{1\text{K}}$ |
| **Use a $1B$ model for probing data** | $\approx 100\text{M}\mathcal{F} + \text{DPO}_{1\text{K}}$ |
| **Self-Instruct** | |
| Synthesize $1K$ instructions from seed | $2048\text{K}\mathcal{F}$ |
| Generate 1 response per instruction | $1024\text{K}\mathcal{F}$ |
| Perform SFT on student | $\text{SFT}_{10\text{K}}$ |
| **Total** | $\approx 70\text{M}\mathcal{F}$ |

Table 8: FLOPs Computation Table for Different Methods

## F  EXPERIMENTS UNDER THE SELF-EVOLVE SETTING

In our primary experiment, we leveraged a teacher model to generate tailored synthetic data aimed at enhancing the capabilities of a different student model. Here, we shift our focus to explore whether **LLMs can harness synthetic data generated by themselves to achieve self-improvement**—a paradigm we term the "Self-Evolve" setting. To investigate this, we adapt our Montessori-Instruct framework by aligning the student model with the teacher model. Starting from an identical checkpoint, the model generates synthetic data for itself, employing influence scores to identify the most beneficial and tailored samples, and subsequently performs Direct Preference Optimization on itself. Notably, the fine-tuning process begins anew from the initial checkpoint, rather than building upon a post-DPO state. We evaluate this paradigm using both Llama3-8B-Instruct, an instruction-tuned model, and Llama3-8B, its pretrained version, to assess the potential of self-improvement. The results are presented in Table 9.

Our findings reveal that Llama3-8B-Instruct achieves superior performance across all benchmarks under the self-evolve setting, exhibiting a consistent upward trend in capability. Remarkably, even the non-instruction-tuned Llama3-8B demonstrates self-improvement at the 8B parameter scale. However, while Llama3-8B exhibits gains with each iteration, the rate of improvement diminishes over time. This suggests that the pretrained model struggles to surpass its instruction-tuned version

Table 9: Self-improvement performance of Llama3 models across different iterations. The Winning Rate (WR) and Length-Control Winning Rate (LC-WR) are compared to the Llama3-8B-Instruct model. The best performances are marked in **bold**.

| Methods | Alpaca Eval 2.0 | | MT-Bench |
|---|---|---|---|
| | WR | LC-WR | Score |
| Llama3-8B-Instruct | 50.00% | 50.00% | 7.472 |
| Llama3-8B (No fine-tuning) | 3.39% | 2.09% | 5.597 |
| **Teacher=Student=**Llama3-8B | | | |
| Iteration 1 | 26.76% | 26.53% | 6.224 |
| Iteration 2 | 35.42% | 34.76% | 6.308 |
| Iteration 3 | 39.84% | 38.12% | 6.386 |
| **Teacher=Student=**Llama3-8B-Instruct | | | |
| Iteration 1 | 53.74% | 52.51% | 7.563 |
| Iteration 2 | 56.78% | 54.84% | 7.595 |
| Iteration 3 | **58.62%** | **56.12%** | **7.611** |

through self-evolution alone at this stage. We attribute this limitation to the suboptimal quality and restricted diversity of the synthetic data produced by the models themselves. Shumailov et al. (2023b) reveals that the perplexity of synthetic training data tends to converge toward a low-value range after multiple iterations, offering diminishing returns in terms of novel and beneficial information for model enhancement. We hope that future research will devise innovative strategies to bridge the gap between synthetic and organic data, unlocking the full potential of self-evolving LLMs.

