# OpenReview forum: "Montessori-Instruct: Generate Influential Training Data Tailored for Student Learning"
_ICLR.cc/2025/Conference — ICLR 2025 Poster_

### Official Review · Reviewer_D2rt · 2024-11-02

**Soundness:** 3
**Presentation:** 3
**Contribution:** 3
**Rating:** 6
**Confidence:** 3

**Summary:**

This paper introduces MONTESSORI-INSTRUCT, a framework to generate synthetic data for training student language models - tailored for the student model's learing process / ability. The method first uses local data influence to measure the utility of synthetic data points for student learning. Then, it optimizes a teacher model with DPO to generate more effective synthetic training data by aligning with the student model's learning preferences.

The authors evaluate MONTESSORI-INSTRUCT using `Llama3-8B-Instruct` as the teacher and `Llama3-8B/Tinyllama-1`.1B as students. Evaluation on Alpaca Eval and MT-Bench shows that Montessori-Instruct outperforms existing methods like Self-Instruct. The authors also show that Montessori-Instruct can beat GPT-4o on in-domain and out-of-domain evaluation. Ablation studies highlight the effectiveness of using data influence to capture student preferences, the benefits of optimizing teacher parameters, and the robustness across different configurations.

**Strengths:**

**Writing**.
- Introduction is well-written. The motivation is clear, the problem is well-defined, key contributions are listed and aligned with the structure of the paper.
- Related work is well-written, extensive, and up-to-date. Related works is also well-organized in logic, and set up a good foundation between existing line of works and the proposed method.

**Evaluation**.
- Evaluation has good coverage of different baselines, and the selection of baselines are realistic. Generally, the evaluation is comprehensive and thorough.
- Ablation study is comprehensive and thorough. The ablation studies the effectiveness of teacher optimization, seed data, (multiple) iterations, and very clearly demonstrates the generalizability of the method.

**Originality**. The idea of the paper is novel, and well-motivated.

**Significance**. The proposed method is a good contribution to the field of synthetic data generation for language model training. The proposed method is generalizable to other domains, though with additional overhead as stated in the limitation section.

**Weaknesses:**

**Scale of experiment.** The limitation section points out that in the experiment, the scale is chosen as a fixed 10k data points. Ablation on the scale of experiment will be helpful to show the generalizability of the method on the scale of data.

**Questions:**

1. Is there an easy way to see how the effectiveness of the framework scales with the data? You are welcome to scale down / scale up, to a point where the experiment is reasonable.

---

> ### Author Response · Authors · 2024-11-21
> **Rebuttal by Authors**
>
> Thank you for your review of our paper! We will address your questions/comments below:
>
> **Question 1**: how does the effectiveness of the framework scale with the data?
>
> **Response**: Thank you for your positive affirmation of our work! The question you raised, which has also been mentioned in our limitations, is indeed a direction for further exploration. In our main experiment, we used 10K data points. To show the benefits of scaling training data, we randomly subsampled 2K, 5K, and 8K data points and further expanded the data volume to 20K. The results are shown in the table below:
>
> | Training Size | LC-WR  | WR     | MT-Bench |
> |---------------|--------|--------|----------|
> | 10K Self-Instruct | 50.00% | 50.00% | 6.490    |
> | 2K Montessori-Instruct | 44.84% | 44.57% | 5.940    |
> | 5K Montessori-Instruct | 50.71% | 51.29% | 6.563    |
> | 8K Montessori-Instruct | 52.32% | 54.49% | 6.785    |
> | 10K Montessori-Instruct | 54.92% | 58.59% | 6.903    |
> | 20K Montessori-Instruct | 55.75% | 59.92% | 6.922    |
>
> We found that when the data size is 5K, the performance of the student surpassed that of the student trained on 10K Self-Instruct data. As the data size increases, we can observe a continuous improvement in the performance of Alpaca Eval and MT-Bench, but the performance grows slower and slower. We believe this is due to two reasons:
>
> 1. As the student is trained, its data preferences will also change, so it is necessary to collect updated data influence to optimize the teacher in order to achieve sustained performance improvement.
>
> 2. As the synthesized data size increases, the ratio of similar data (identified by Rouge-L > 0.7) will increase, and the ratio of useful data will decrease. For example, when the size is 5K, the amount of similar data is ~2K; when the size is 20K, the amount of similar data is ~10K; and when the size is 50K, the amount of similar data reaches ~35K.  **We found this high data duplication rate in all  the baselines**, and we did not find papers studying this phenomenon. We believe this is a challenging issue and will raise this in the discussions of our paper's next version as a call for action for the community. In the future, we will explore diversifying the seed data to alleviate this phenomenon.

---

> > ### Comment · Reviewer_D2rt · 2024-11-26
> > **Thank you**
> >
> > Thank you for you response! I think the comment addresses my concern about the methodology.

---

> > > ### Author Response · Authors · 2024-11-28
> > >
> > > Thank you very much for your response and recognition of our work! If you have any further questions, please don't hesitate to let us know.

---

> ### Author Response · Authors · 2024-11-25
> **Looking Forward to Your Reply**
>
> Dear Reviewer `D2rt`,
>
> We have carefully addressed your feedback in our rebuttals and provided detailed responses to each of your comments, particularly regarding the scale of the experiments. We believe these clarifications will aid in assessing our work more comprehensively.
>
> We would greatly appreciate it if you could review our rebuttals and provide any further feedback, given that the author-reviewer discussion will be closed on Nov. 26 at 11:59 p.m. AoE in no more than two days. We are willing to answer any further questions.
>
> Thank you for your time and consideration. We look forward to your reply.
>
> Best,
>
> The Authors

---

### Official Review · Reviewer_PpzE · 2024-11-04

**Soundness:** 3
**Presentation:** 3
**Contribution:** 4
**Rating:** 8
**Confidence:** 4

**Summary:**

This paper introduces Montessori-Instruct, a novel framework that optimizes the teacher model's data generation capabilities by aligning them with student learning preferences. The framework uses local data influence measurements to estimate how synthetic data impacts student learning, then employs DPO to tune the teacher model accordingly. Experiments with Llama-3 and TinyLlama models showed significant improvements over baseline methods.

**Strengths:**

- A novel data influence-driven approach where the teacher model actively learns to generate training data by optimizing against student's learning preferences, in contrast to previous works that use a static teacher.
- The experimental comparisons and benchmarks are comprehensive, the proposed method shows significant improvements, and the ablation and analysis experiments are thorough.

**Weaknesses:**

The main drawback of this work is the excessive computational overhead beyond the objective of training the student model. Although the authors discussed this additional computation in detail in Section 6 and Appendix E, I believe that the impact on the paper's practicality cannot be dismissed through discussion alone. The majority of the additional computational cost comes from local data influence for the student, where one step of training is required for each sample, along with evaluation on the reference dataset. To accelerate this process, the authors used 8xH100, which is a very expensive overhead. This makes me question whether we could actually achieve similar gains by investing these resources in using stronger teacher models and building more sophisticated reward model pipelines[1]. I encourage the authors to (1) discuss potential directions that could directly reduce the unit cost of local data influence, rather than encouraging more computational resources, thereby enhancing the promise of local data influence-based approaches. (2) present the additional monetary costs for each method (i.e., APIs used for baseline methods and computational resources used for acceleration) to improve the fairness and transparency of the comparison.

[1] Snell, Charlie, et al. "Scaling llm test-time compute optimally can be more effective than scaling model parameters." _arXiv preprint arXiv:2408.03314_ (2024).

**Questions:**

- In lines 226-228, how were the 6,792 preference pairs collected from the 10K probing dataset?
- Will the general capabilities of the optimized teacher model deteriorate?

---

> ### Author Response · Authors · 2024-11-21
> **Rebuttal by Authors Part 1**
>
> Thank you for your review of our paper! We will address your questions/comments below:
>
> **Weakness**: concerns about the additional cost
>
> **Response**:
>
> **(1) 8xH100 is a very expensive overhead**
>
> We used the H100 to complete the experiments on the main table, which utilized the local data influence of 10K data to optimize the teacher, solely to achieve the best performance. As we mentioned in the general response, we only need about 1K local data influences to enable the student’s performance to surpass other baselines, which only require 8 A6000s or 4 A100-80GBs and can be finished in 3 hours. This actually provides researchers with diverse options: if resources are limited, generating approximately 1,000 data influences to optimize the teacher can yield significantly superior performance (2.40% for LC-WR, 5.85% for WR, and 0.304 for MT-Bench), all of which outperform the baselines. If resources are sufficient, the teacher can be continuously optimized to further raise the performance ceiling.
>
> **(2) whether we could achieve similar gains by investing these resources in using stronger teacher models**
>
> We believe that a strong teacher does not necessarily mean that it can generate equally high-quality synthetic data, as mentioned in this paper[1]. Thus, targeted optimization is an important condition for synthesizing high-quality data. We represent another dimension of improvement, one that not only enhances teacher’s capability but also supports student’s learning. Moreover, the construction of complex reward models is also very complicated and resource-intensive.
>
> In fact, the resources we invest in optimizing the teacher represent a one-off cost. In Section 5.4, we fine-tuned four different models, including Qwen, Gemma, and Mistral, using the data synthesized by the same optimized teacher, all of which achieved better performance than the baseline. Therefore, our method allows the teacher to be trained once and fixed, without the need to retrain every time a student is changed.
>
> **(3) Discuss potential directions that could reduce the unit cost**
>
> We believe the most direct approach is to use a proxy model, as demonstrated in our experiments, where the local data influence obtained from a 1B small model can generalize well to an 8B model (54.13%WR, 53.61%LC-WR and 6.83 for MT-Bench). This can help reduce our computational load to a level comparable to the Self-Instruct method. Another future direction is to utilize classifier models, such as BERT-based models, to further accelerate the process of obtaining data influence[1]. Thank you for your suggestion.
>
> **(4) present the additional monetary costs for each method**
>
> Thank you for your suggestion! We calculated the FLOPs of each method in the General Response. According to the results, the FLOPs of our method is lower than both the Self-Reward and LLM2LLM baselines when using the proxy model, and it is already very close to Self-Instruct. However, our method demonstrates a significant improvement for the student, with increases of 3.61% for LC-WR, 4.13% for WR, and 0.34 for MT-Bench. Regarding the monetary costs, the A6000 is priced at \\$1 per hour, the A100 at \\$1.50 per hour, and the H100 at \\$2 per hour. In the Self-Instruct and LLM2LLM methods, we utilize GPT-4o to generate synthetic data. The API cost for GPT-4o is \\$2.50 per 1 million input tokens and \\$10.00 per 1 million output tokens. Consequently, the average cost for synthesizing 10,000 data points (considering data waste) amounts to approximately \\$53.20, which is significantly more expensive than using GPUs. So in fact, our method is superior in terms of both computational load and cost.

---

> ### Author Response · Authors · 2024-11-21
> **Rebuttal by Authors Part 2**
>
> **Question 1**: In lines 226-228, how were the 6,792 preference pairs collected from the 10K probing dataset?
>
> **Response**: Thank you for pointing it out!  As described in Section 3.3 (lines 196–199), we create preference pairs that satisfy the following conditions: 1) they share the same seed prompt, and 2) one has a positive influence while the other has a negative influence. To achieve this, we first aggregate the data in the 10K probing dataset by their seed prompts, dividing them into positive and negative groups. Every time we select one data point from the positive group and one from the negative group, if the two data points are generated from a common seed prompt, we combine them into a preference data pair. Since some seed prompts generate data with only positive or negative influences, we end up with 6,792 pairs from the 10K probing dataset.
>
> **Question 2**: Will the general capabilities of the optimized teacher model deteriorate?
>
> **Response**: Thanks for raising this invaluable question! We conducted comprehensive testing on the Llama3-8B-Instruct teacher (with Llama3-8B as the student) before and after DPO on MT-Bench, MMLU, GSM8K, GPQA, ARC-C, and HellaSwag. The results are as follows:
>
> |                    |  MT-Bench |  MMLU |  GPQA | ARC-C | GSM8K | HellaSwag |
> |:------------------:|:--------:|:-----:|:-----:|:-----:|:-----:|:---------:|
> | Llama3-8B-Instruct | 7.472  | 66.21 | 31.96 | 59.54 | 73.48 |   77.21   |
> |  Teacher-DPO-Iter1 | 7.473  | 65.95 | 31.72 | 59.15 | 73.65 |   76.86   |
> |  Teacher-DPO-Iter2 | 7.465  | 66.07 | 32.54 | 58.86 | 73.14 |   77.08   |
>
> The results indicate that the teacher's ability, after specific optimization in data synthesis capability, is basically on par with the original model with minimal fluctuations, demonstrating that optimizing the teacher's data synthesis capability does not adversely affect performance on OOD tasks.
>
> [1]: Yu, Z., Das, S., & Xiong, C. (2024). MATES: Model-Aware Data Selection for Efficient Pretraining with Data Influence Models. *arXiv preprint arXiv:2406.06046*.

---

> > ### Comment · Reviewer_PpzE · 2024-11-25
> >
> > Thank you for your comprehensive reply! The small model proxy is indeed a good point. My concerns have been largely addressed.
> >
> > I sincerely believe this work represents one of the general solutions to the current problem that "stronger teachers may not necessarily be better at teaching students," if its computational efficiency can be effectively improved in the future. The data influence calculation faithfully reflects the samples' contributions to real downstream tasks and can be used to adjust the teacher model.
> >
> > Therefore, I am inclined to accept this work.

---

> > > ### Author Response · Authors · 2024-11-25
> > >
> > > Thank you very much for your response and recognition of our work! We will add the assessment results of teachers' general abilities and further explain how to use the small proxy model in the next version.

---

### Official Review · Reviewer_5Czb · 2024-11-04

**Soundness:** 3
**Presentation:** 3
**Contribution:** 2
**Rating:** 6
**Confidence:** 3

**Summary:**

This paper proposes a customized synthetic data generation pipeline that seeks to improve the expert model towards generating more useful and debiased data for the student model by aligning the expert model towards data distribution with higher influences on the student model via DPO. Specifically, this paper (1) first adopts an established influence function in active learning to measure the utility of a single data point in the probing set, (2) then they construct preference data using the positive and negative data influences sampled from a same prompt, and use this data to align the expert model. Finally, they regenerate data from the updated expert model and finetune the student model with this data. While the method proposed in this paper generally make sense, it suffers from a lack of ablation study and analysis on the additional cost (e.g. manual curation cost of the reference set, per sample influence inference cost, training cost of DPO on expert model) incurred by this data selection process.

**Strengths:**

This paper proposes a synthetic data generation pipeline to customizedly align the expert model towards higher data utility over the student model. Specifically,
+ the method proposed is sound and can intuitively address the OOD issue with the synthetic data to enhance the generalizability of the student model.
+ this paper has good presentations and conveys the idea clearly.

**Weaknesses:**

The paper lacks more informative ablation study and in-depth explanation and analysis, especially,
+ How do the authors pick the reference dataset and determine its size? It seems this dataset is critical to judge the influence of the data samples on the student model and highly determine the performance of the fine-tuned student model later.
+ Can the author provide some ablation study on the size of the reference dataset and its selection method?
+ How many samples do the authors obtain for each prompt in order to get a pair of positive and negative instruction? What are the additional costs behind these?
+ Calculating the inference for each data point in the probing set seems a bit costly as it involves an LLM optimization (even if just one step). Can the authors provide an ablation study on the sample size of the probing set and the final performance?
+ It also introduces some other additional costs as the proposed method involves DPO on the expert model, while most of the baselines do not. As the expert model is large in size, this training cost seems also non-negligible.

**Questions:**

See the weakness section.

---

> ### Author Response · Authors · 2024-11-21
> **Rebuttal by Authors Part 1**
>
> Thank you for your review of our paper! We will address your questions/comments below:
>
> **Weakness 1**: How to pick the reference dataset and determine its size?
>
> **Response**: Thanks for raising this question! Regarding the choice of the reference dataset, we have two guiding principles: First and foremost, we select reference tasks that reflect the target capabilities we want the LLM to achieve. Following the practices of some previously accepted excellent works [1][2][3], we chose to use in-domain data that is the same as the seed data, specifically alpaca gpt4[4]. This is a dataset synthesized by GPT4 using prompts in the Alpaca format, aimed at improving the model's instruction-following ability. Second, we ensure that there is no data leakage to prevent potential overfitting. Our OOD experiments show good generalization ability of students, which demonstrates that we are not overfitting the targeted task. Regarding the size of the reference dataset, we selected the best quantity that balances performance and efficiency within the limits of our computational resources, which is 256.
>
> We conducted ablation experiments on different reference datasets and different reference dataset sizes.
>
> Regarding the size of the reference dataset, in addition to the original 256, we also chose 8, 32, and 128 for experimentation.
>
> | Size | LC-WR  | WR     | MT-Bench | Correlation |
> |------|--------|--------|----------|----------|
> | 8    | 56.74% | 60.23% | 6.672    | 0.940 |
> | 32   | 54.30% | 59.53% | 6.654    | 0.992 |
> | 128  | 53.29% | 56.70% | 6.820    | 0.984 |
> | 256  | 54.92% | 58.59% | 6.903    | 1.000 |
>
> The experimental results show that changing the reference dataset size has a small impact on the student's performance. Although it performs well on Alpaca Eval when the size is 8, it performs poorly on MT-Bench, which may be due to the randomness of selecting the 8 data points. We also calculated the correlation coefficients between the data influence generated by different sizes and the data influence when the size is 256. All the correlation coefficients are greater than 0.9, demonstrating a very strong correlation among the reference datasets of different sizes. Therefore, the chosen size of 256 can achieve the best overall performance on both Alpaca Eval and MT-Bench metrics.
>
> In addition to the original alpaca gpt4 as the reference dataset, we chose two other datasets: DOLLY[5] and Open Assistant[6]. These two open-ended generation datasets with human-written answers contain various forms of data, while the answers in the alpaca gpt4 dataset are synthesized by the model rather than written by humans. The results are shown below:
>
> | Seed data      | LC-WR  | WR     | MT-Bench |
> |----------------|--------|--------|----------|
> | Alpaca GPT4    | 54.92% | 58.59% | 6.903    |
> | Dolly          | 53.77% | 54.62% | 6.752    |
> | Open Assistant | 48.76% | 51.48% | 6.946    |
>
> Using Dolly and Open Assistant as reference datasets will lower the scores of Alpaca Eval. We believe this decrease comes from that Dolly and Open Assistant are responses provided by humans, while Alpaca GPT4 generates answers using the GPT4 model. The latter has a smaller distribution difference from the model itself, making it easier to learn from. Additionally, we believe that the reason Open Assistant improves the MT-Bench score is that MT-Bench measures the model's ability in multi-turn dialogue, and only Open Assistant among these three reference datasets contains multi-turn dialogue data. Overall, our method demonstrates robustness on reasonable reference tasks.
>
> **Weakness 2**: More details about preference data pairs
>
> **Response**: We generate four instructions for each prompt, with each instruction assigned to a different local data influence. From these four instructions, we select one with positive influence and one with negative influence to form a preference data pair.  If there are multiple positive/negative data influences, we will pair them together randomly, resulting in more than one preference data point for a single prompt. We will include more details in the next version of the paper.

---

> ### Author Response · Authors · 2024-11-21
> **Rebuttal by Authors Part 2**
>
> **Weakness 3**: Ablation study on the sample size of the probing set and the final performance
>
> **Response**: Thank you for raising this question. We actually conducted this experiment in the original paper and illustrated the relationship between student performance and teacher training steps in Figures 3(a) and 3(b), where the size of the probing dataset determines the teacher training steps since the teacher is trained with one epoch. To clarify, we used "probing dataset size" as the x-axis in Figures 3(a) and 3(b) instead of the teacher training steps. The discrete data points used to plot the relationship between probing dataset size and student performance are shown in the table below:
>
> | Probing dataset size | LC-WR  | WR     | MT-Bench |
> |----------------------|--------|--------|----------|
> | 500                  | 51.86% | 53.63% | 6.652    |
> | 1000                 | 52.40% | 55.85% | 6.794    |
> | 2000                 | 53.77% | 58.05% | 6.681    |
> | 3000                 | 54.42% | 57.81% | 6.852    |
> | 5000                 | 54.10% | 58.20% | 6.920    |
>
> As shown, when the probing dataset size reaches 1K, the student's performance already surpasses the baseline. As mentioned in the general response, if researchers do not have sufficient resources, they can choose to use 1K probing data, which will ensure that the performance of the student model achieves the best results compared to other baselines while maintaining efficiency. We generated 6,792 probing data for the main experiments in our paper and achieved overall optimal performance within the resources available to us.
>
> **Weakness 4**:  some other additional costs as the proposed method involves DPO on the expert model, while most of the baselines do not.
>
> **Response**:  We want to clarify that the other two baselines—LLM2LLM and Self-Reward—also require additional resources: LLM2LLM relies on larger models via API calls to generate instructions, while Self-Reward introduces a separate 70B expert model dedicated solely to generating instructions for students. In contrast, our method requires only an additional DPO step for the teacher. The FLOPs of DPO can be estimated as 4 times the Policy Model Forward FLOPs plus 2 times the Reward Model Forward FLOPs, while the FLOPs of SFT can be estimated as 3 times the Policy Model Forward FLOPs. Although the FLOPs of DPO are higher than those of SFT on a per-unit data basis, DPO does not incur much additional consumption because it uses a smaller total amount of data. According to the Performance-FLOPs relationship in the General Response, even if we introduce DPO for the teacher, our overall FLOPs are still comparable to Self-Instruction.
>
> This DPO process can be beneficial, as verified in a recent paper[7], which aligns with our experimental findings: a strong model does not necessarily excel at synthesizing high-quality data. Therefore, even with a strong teacher, producing high-quality data may still require investing additional resources to refine the teacher.
>
> [1]: Xia, M., Malladi, S., Gururangan, S., Arora, S., & Chen, D. (2024). Less: Selecting influential data for targeted instruction tuning. *arXiv preprint arXiv:2402.04333*.
>
> [2]: Paul, M., Ganguli, S., & Dziugaite, G. (2021). Deep learning on a data diet: Finding important examples early in training. *Advances in neural information processing systems, 34, 20596–20607*.
>
> [3]: Xiaobo Xia, Jiale Liu, Jun Yu, Xu Shen, Bo Han, & Tongliang Liu (2023). Moderate Coreset: A Universal Method of Data Selection for Real-world Data-efficient Deep Learning. *In The Eleventh International Conference on Learning Representations*.
>
> [4]: Peng, B., Li, C., He, P., Galley, M., & Gao, J. (2023). Instruction Tuning with GPT-4. *arXiv preprint arXiv:2304.03277*.
>
> [5]: Conover, M., Hayes, M., Mathur, A., Xie, J., Wan, J., Shah, S., Ghodsi, A., Wendell, P., Zaharia, M., and Xin, R. Free. Dolly: Introducing the world’s first truly open instruction-tuned LLM, 2023.
>
> [6]: Kopf, A., et al. "Openassistant conversations-democratizing large language model alignment," in *Advances in Neural Information Processing Systems, vol. 36, 2024*.
>
> [7]: Xu, Z., Jiang, F., Niu, L., Lin, B. Y., & Poovendran, R. (2024). Stronger Models are NOT Stronger Teachers for Instruction Tuning. *arXiv preprint arXiv:2411.07133*.

---

> ### Author Response · Authors · 2024-11-25
> **Looking Forward to Your Reply**
>
> Dear Reviewer `5Czb`,
>
> We have carefully addressed your feedback in our rebuttals and provided detailed responses to each of your comments, particularly regarding the ablation studies on the reference dataset, the size of the probing dataset, and a thorough analysis of our cost-performance relationship. We believe these clarifications will aid in assessing our work more comprehensively.
>
> We would greatly appreciate it if you could review our rebuttals and provide any further feedback, given that the author-reviewer discussion will be closed on Nov. 26 at 11:59 p.m. AoE in no more than two days. We are willing to answer any further questions.
>
> Thank you for your time and consideration. We look forward to your reply.
>
> Best,
>
> The Authors

---

> > ### Comment · Reviewer_5Czb · 2024-11-26
> >
> > I thank the authors for their detailed response, which has addressed most of the concerns. Regarding the response to W1, could the authors provide more analysis on why simply using in-domain data can significantly boost OOD performance using their method? Since it appears that their method selects samples simply based on their influence over the reference data, which has no implication on the distribution of the test set. Regardless, I have improved my score.

---

> > > ### Author Response · Authors · 2024-11-28
> > >
> > > Thank you for your response and for your additional questions! We believe the reason lies in two aspects:
> > >
> > > 1. Montessori-Instruct takes student preferences into account when generating synthetic data. The pre-trained model has already acquired sufficient knowledge[1], while instruction tuning is not intended to inject new knowledge but rather to align the query (instruction) with the model's internal knowledge[2][3]. This alignment enhances the model's ability to utilize its existing knowledge, thereby improving its performance on both in-domain and out-of-domain tasks. Specifically, our method calculates the data influence scores of various instructions on the reference dataset. A higher score indicates that this data is more beneficial for the model in aligning external queries with its inherent knowledge, thereby enhancing the model's ability to leverage its internal knowledge.
> > >
> > > 2. Montessori-Instruct utilizes in-domain data as a reference to guide the generation of training data, but it does not train directly on the in-domain data. The actual training data is generated by the teacher and may contain general information that enhances the capabilities of the LLMs. We believe that this can lead to an effective fine-tuning stage, thereby improving performance on out-of-domain tasks[4][5].
> > >
> > > If you have any further questions, please don't hesitate to let us know. Thank you for acknowledging our work.
> > >
> > > [1] Chunting Zhou, Pengfei Liu, Puxin Xu, Srini Iyer, Jiao Sun, Yuning Mao, Xuezhe Ma, Avia Efrat, Ping Yu, LILI YU, Susan Zhang, Gargi Ghosh, Mike Lewis, Luke Zettlemoyer, & Omer Levy (2023). LIMA: Less Is More for Alignment. *In Thirty-seventh Conference on Neural Information Processing Systems.*
> > >
> > > [2] Mengjie Ren, Boxi Cao, Hongyu Lin, Cao Liu, Xianpei Han, Ke Zeng, Wan Guanglu, Xunliang Cai, and Le Sun (2024). Learning or Self-aligning? Rethinking Instruction Fine-tuning. *In Proceedings of the 62nd Annual Meeting of the Association for Computational Linguistics (Volume 1: Long Papers)*, pages 6090–6105, Bangkok, Thailand. Association for Computational Linguistics.
> > >
> > > [3] Wei Liu, Weihao Zeng, Keqing He, Yong Jiang, & Junxian He (2024). What Makes Good Data for Alignment? A Comprehensive Study of Automatic Data Selection in Instruction Tuning. *In The Twelfth International Conference on Learning Representations.*
> > >
> > > [4] Wei, J., Bosma, M., Zhao, V., Guu, K., Yu, A., Lester, B., Du, N., Dai, A., & Le, Q. (2022). Finetuned Language Models are Zero-Shot Learners. *In International Conference on Learning Representations.*
> > >
> > > [5] Nihal Nayak, Yiyang Nan, Avi Trost, and Stephen Bach. 2024. Learning to Generate Instruction Tuning Datasets for Zero-Shot Task Adaptation. *In Findings of the Association for Computational Linguistics: ACL 2024, pages 12585–12611*, Bangkok, Thailand. Association for Computational Linguistics.

---

> > > > ### Comment · Reviewer_5Czb · 2024-12-02
> > > > **Thank you for your response!**
> > > >
> > > > Thank you for the response, which has addressed most of my concerns. I have further improved the score.

---

> > > > > ### Author Response · Authors · 2024-12-02
> > > > >
> > > > > Thank you very much for your response and recognition of our work! If you have any further questions, please don't hesitate to let us know.
> > > > >
> > > > > Best,
> > > > >
> > > > > The Authors

---

### Official Review · Reviewer_hB25 · 2024-11-04

**Soundness:** 3
**Presentation:** 3
**Contribution:** 2
**Rating:** 6
**Confidence:** 3

**Summary:**

This paper introduces Montessori-Instruct, a new framework for generating synthetic training data to enhance the learning of student language models. The authors propose a method that aligns data synthesis by a teacher model (e.g., Llama3-8B-Instruct) with the learning process of the student model (e.g., Llama3-8B). The approach involves assessing the local data influence of synthetic data points on the student model to understand its learning preferences, followed by training the teacher model using Direct Preference Optimization (DPO) to generate data better suited for the student. Experiments show that Montessori-Instruct significantly improves performance on benchmarks like Alpaca Eval and MT-Bench.

**Strengths:**

* The concept of aligning the teacher model to better match the student model’s learning preferences is well-motivated.

* The method of assessing the impact of individual data points on student learning (i.e., local data influence) is a interesting approach backed by theoretical guarantees.

**Weaknesses:**

* Including experiments with multiple runs and reporting the mean and standard deviation would strengthen the reliability of the results.

* The performance gains, particularly on out-of-domain benchmarks, appear minimal when weighed against the additional computational cost involved in training the teacher model.

* The idea of local data influence, which is the crucial algorithm for the proposed solution, is not new but taken from a previous paper, as the authors have mentioned. This reliance on existing techniques may reduce the perceived originality of the paper.

**Questions:**

See weaknesses above

---

> ### Author Response · Authors · 2024-11-21
> **Rebuttal by Authors**
>
> Thank you for your review of our paper! We will address your questions/comments below:
>
> **Weakness 1**: Include experiments with multiple runs and report the mean and the standard deviation.
>
> **Response**: Thank you for pointing it out! We conducted three repeated experiments on the Llama3-8B group using different seeds. We report the mean and standard deviation of the evaluation results.
>
> |                    | LC-WR                | WR                   | MT-Bench          | MMLU               | GPQA               | ARC-C              | GSM8K              | HellaSwag          |
> |--------------------|----------------------|----------------------|-------------------|--------------------|--------------------|--------------------|--------------------|--------------------|
> | Llama3-8B-iter1    | 54.88% ± 0.39% | 58.51% ± 0.35% | 6.90 ± 0.05 | 62.78 ± 0.15 | 30.06 ± 0.31 | 62.32 ± 0.78 | 58.47 ± 0.57 | 81.10 ± 0.14 |
> | Llama3-8B-iter2    | 56.81% ± 0.39% | 60.55% ± 0.36% | 7.18 ± 0.06 | 63.32 ± 0.21 | 31.86 ± 0.43 | 60.40 ± 0.31 | 60.03 ± 0.39 | 81.90 ± 0.14 |
>
>
> As shown in the table, our method demonstrates strong stability on both the Alpaca Eval and OOD benchmarks. Even in the case of  $-3\sigma$, we are still better than the baseline. While the fluctuations in MT-Bench are slightly larger compared to the other benchmarks, we attribute this to the inherent instability of using LLM-as-a-judge in MT-Bench. We will repeat the other experiments in the following time and report the mean and standard deviation of all methods in the final version.
>
> **Weakness 2**:  The performance gain on OOD benchmarks appears minimal
>
> **Response**: Thank you for raising this question! We want to clarify that our primary goal is to enhance the model’s instruction-following ability (evaluated using Alpaca Eval for in-domain and MT-Bench for OOD) while also preserving its generalization ability across other tasks (e.g., Math, QA). Synthetic data often exacerbate OOD issues, as evidenced by performance declines on OOD benchmarks for our baselines (e.g., a 3.73 point drop for Self-Reward-iter1 and a 2.8 point drop for LLM2LLM-iter1 on GPQA). However, our method effectively addresses this challenge by maintaining strong overall performance on OOD benchmarks, with notable improvements of 0.673 points on MT-Bench, 1.05 points on MMLU, 1.26 points on GSM8K, 2.99 points on ARC-C, and 1.05 points on HellaSwag (you can find more details in Table 1 in our paper). This strength was also highlighted by Reviewer 5Czb, who noted: “***the method proposed is sound and can intuitively address the OOD issue with the synthetic data to enhance the generalizability of the student model.***” We will emphasize this further in the next version of the paper.
>
> **Weakness 3**: This reliance on existing techniques may reduce the perceived originality of the paper.
>
> **Response**: We want to clarify that our focus is not on emphasizing local data influence as an innovation point. Influence functions are a well-established method in statistics and have a wide range of applications[1][2][3][4], except for guiding the generation of synthetic data. Our contribution lies in being the first to leverage data influence to guide the synthetic data generation process and demonstrate its superiority, as highlighted in lines 85-86 in the paper: “We incorporate influence functions to accurately capture the student’s data preferences and effectively guide the teacher’s optimization directions.” Also, in the ablation experiment titled “Effectiveness of Local Data Influence” in Section 5.3, we compared the performance of IF and an LLM-judger in evaluating data quality. The results clearly show that IF outperforms the LLM-as-a-judge, with a 1.50% improvement on LC-WR, a 3.66% improvement on WR, and a 0.172 point gain on MT-Bench. This shows the effect of introducing IF. We believe that incorporating IF in the generation of synthetic data has huge potential also in pre-training and post-training, given the importance and wide usage of synthetic data.
>
> [1]: Sanford Weisberg and R Dennis Cook. Residuals and influence in regression. 1982.
>
> [2]: Koh, P., & Liang, P. (2017). Understanding black-box predictions via influence functions. *In International conference on machine learning (pp. 1885–1894)*.
>
> [3]: Park, S., Georgiev, K., Ilyas, A., Leclerc, G., & Madry, A. (2023). Trak: Attributing model behavior at scale. *arXiv preprint arXiv:2303.14186*.
>
> [4]: Grosse, R., Bae, J., Anil, C., Elhage, N., Tamkin, A., Tajdini, A., Steiner, B., Li, D., Durmus, E., Perez, E., & others (2023). Studying large language model generalization with influence functions. *arXiv preprint arXiv:2308.03296*.

---

> ### Author Response · Authors · 2024-11-25
> **Looking Forward to Your Reply**
>
> Dear Reviewer `hB25`,
>
> We have carefully addressed your feedback in our rebuttals and provided detailed responses to each of your comments, particularly regarding the experiments involving multiple runs and the further explanation of our OOD performance gains. We believe these clarifications will enhance the comprehensive assessment of our work.
>
> We would greatly appreciate it if you could review our rebuttals and provide any further feedback, given that the author-reviewer discussion will be closed on Nov. 26 at 11:59 p.m. AoE in no more than two days. We are willing to answer any further questions.
>
> Thank you for your time and consideration. We look forward to your reply.
>
> Best,
>
> The Authors

---

> ### Author Response · Authors · 2024-12-02
> **Looking Forward to Your Reply**
>
> Dear Reviewer `hB25`,
>
> Given that Dec 2nd is the last day for reviewers to post messages, we would greatly appreciate it if you could review our rebuttals and provide any further feedback. We are willing to answer any further questions.
>
> Thank you for your time and consideration.
>
> Best,
>
> The Authors

---

> > ### Comment · Reviewer_hB25 · 2024-12-02
> >
> > Thanks for the detailed response. The response has resolved my concerns, and I have updated the score accordingly.

---

> > > ### Author Response · Authors · 2024-12-02
> > >
> > > Thank you very much for your response and recognition of our work! If you have any further questions, please don't hesitate to let us know.
> > >
> > > Best,
> > >
> > > The Authors

---

### Author Response · Authors · 2024-11-21
**General Rebuttal**

# 0 Overview

We thank all the reviewers for their great efforts.

**In this post**:
1. We summarize the positive points that the reviewers unanimously agree on.
2. We provide further clarification on the cost-performance relationship of our method compared to all baselines, along with a discussion on how the FLOPs utilized during post-training compare with those used during pretraining and test-time scaling. While the reviewers expressed concerns regarding the cost of our approach, our analysis demonstrates that Montessori-Instruct achieves better results at a cost comparable to Self-Instruct and is better than the best performance achieved by Self-Reward and LLM2LLM.
3. We provide the results of applying our method to self-play, where the student can also serve as the teacher to improve itself.

In the individual replies, we address other comments. We have also added a new appendix section to the paper, which is highlighted in red. The updated PDF has been re-uploaded.

# 1 Positive statements

- We sincerely thank the reviewers for recognizing the key contributions of our work. We appreciate their acknowledgment of Montessori-Instruct as a well-motivated and novel pipeline, particularly its use of data influence as rewards to align the teacher’s generation with the student’s preferences. (`hB25`, `5Czb`, `PpzE`, `D2rt`).
- We are grateful for their recognition of our method’s strong generalization ability in addressing OOD challenges (`5Czb`, `D2rt`), its theoretical guarantees, dynamic teacher design, and superior performance demonstrated through ablation studies (`D2rt`, `hB25`, `PpzE`).
- We thank the reviewers for praising our clear and well-written paper (`hB25`, `5Czb`, `PpzE`, `D2rt`).

---

> ### Author Response · Authors · 2024-11-21
> **2 Cost-performance relation of all the methods**
>
> # 2 Cost-performance relation of all the methods
> We further explain the reviewers' questions about the consumption of our method. We analyzed the **Performance-FLOPs curve** of four methods, with a particular focus on the changes in Self-Instruct's Alpaca Eval and MT-Bench Score as their FLOPs increase to levels comparable to those of Montessori-Instruct. We scale the FLOPs of Self-Instruct by synthesizing additional data. We also marked the Performance-FLOPs relationship of the two baselines, LLM2LLM and Self-Reward, in **Figure 15 (a), Figure 15 (b) and Figure 15 (c)** . We have also attached the PDF versions of these three figures in the uploaded files.
>
> According to the figures, it can be seen that Self-Instruct quickly reached the upper bound during the scaling-up process, and even with more FLOPs, no better performance improvement can be achieved. The reason may be that the data generated by Self-Instruct is severely homogenized. In contrast, the upper bound of our method is significantly better and continuously grows when we invest more FLOPs into it.
>
> Then we give a computational result of the FLOPs estimated for four methods, as well as the pretraining and test-time-scaling. The main FLOPs for Montessori-Instruct come from processing probing data. In the main table, we used 10K probing data to utilize the most resources to achieve the best performance, but as the Figure 3(a) and 3(b) in our paper suggests, using ~1K probing data can already achieve better performance than other baselines.  To make a fair comparison, we calculate the FLOPs under 1K probing data. We estimate the FLOPs as follows (Llama3-8B-Instruct as the teacher, Llama3-8B as the student):
>
> - Self-Instruct: $1.34\times10^{20}$ FLOPs
> - Self-Reward: $2.11\times10^{21}$ FLOPs
> - LLM2LLM: $2.3\times10^{20}$ FLOPs
> - Montessori-Instruct: $6.43\times10^{20}$ FLOPs
> - Pretrain Llama3-8B: $1.87\times10^{24}$ FLOPs
> - Inference-Time Scaling : $1.60\times10^{23}$ FLOPs
>
> We can see that Montessori-Instruct's FLOPs are 7 times less than Self-Reward, which is the current SOTA method. Furthermore, if we use the proxy model[1], such as a smaller-sized model (e.g., 1B parameters for assisting an 8B model) to process probing data, Montessori's FLOPs can further reduce to $1.92\times10^{20}$ FLOPs. This makes it comparable to Self-Instruct while still outperforming it. Using a proxy model has promising potential for enhancing both efficiency and performance, which we leave for future work. Regarding the pretraining, since the computational cost during the SFT phase is significantly lower than that during the pretraining phase ( $10^4$ times smaller), even if we increase resource investment in SFT, its overall consumption remains minimal. Recent work has focused on scaling inference time to achieve better performance [2]. However, the inference-time scaling FLOPs are also significantly larger than those of SFT, being approximately $10^3$ times greater, according to [3]. Nevertheless, our teacher training represents a one-time cost. As demonstrated in Section 5.4 of the paper, the optimized teacher can assist multiple students in improving their performance without the need for retraining from scratch.
>
> The detailed derivation is provided in Section E.3 of the new version of the paper in the Appendix.
>
>
>
> [1]: Yu, Z., Das, S., & Xiong, C. (2024). MATES: Model-Aware Data Selection for Efficient Pretraining with Data Influence Models. *arXiv preprint arXiv:2406.06046*.
>
> [2]: Snell, C., Lee, J., Xu, K., & Kumar, A. (2024). Scaling llm test-time compute optimally can be more effective than scaling model parameters. *arXiv preprint arXiv:2408.03314*.
>
> [3]: Sardana, N., Portes, J., Doubov, S., & Frankle, J. (2023). Beyond chinchilla-optimal: Accounting for inference in language model scaling laws. *arXiv preprint arXiv:2401.00448*.

---

> > ### Author Response · Authors · 2024-11-21
> > **3 Self-play experiments**
> >
> > # 3 Self-play experiments
> > We use a teacher-student framework in the main paper to ensure a fair comparison with other baselines, as they all rely on a strong model to generate instructions for the student. However, our method also demonstrates even greater potential in the self-play setting, where the same model serves as both teacher and student. We conducted the experiments under the same conditions outlined in Table 1, using LLama3-8B-Instruct as both the teacher and the student, which yielded promising results.
> >
> > |                          | Alpaca Eval WR | Alpaca Eval LC-WR | MT-Bench |
> > |--------------------------|----------------|--------------------|----------|
> > | Llama3-8B-Instruct       | 50.00%         | 50.0%             | 7.472    |
> > | Llama3-8B-Instruct-iter1 | 53.74%         | 52.51%            | 7.563    |
> > | Llama3-8B-Instruct-iter2 | 56.78%         | 54.84%            | 7.595    |
> > | Llama3-8B-Instruct-iter3 | 58.62%         | 56.12%            | 7.611    |
> >
> > Llama3-8B-Instruct shows continuous growth on both the in-domain Alpaca Eval and the out-of-domain MT-Bench, demonstrating the exciting prospects brought by our method combined with self-play.

---

### Meta-Review · Area_Chair_FZiN · 2024-12-23

**Metareview:**

This paper proposes to use influence function to select training examples in the iterative instruction tuning process. Despite increasing the computation cost due to evaluating the influence function, it can outperform other iterative improvement methods and the observations are valuable for future studies.

**Additional Comments On Reviewer Discussion:**

The reviewers raised concerns on the FLOPs comparison with other methods and the authors gave the details.

---

### Decision · Program_Chairs · 2025-01-22

Accept (Poster)